# FACE: A General Framework for Mapping Collaborative Filtering Embeddings into LLM Tokens

**Chao Wang**[1][†][*]  **Yixin Song**[1][†]  **Jinhui Ye**[2]  **Chuan Qin**[3]
**Dazhong Shen**[4]  **Lingfeng Liu**[1]  **Xiang Wang**[1]  **Yanyong Zhang**[1]

[1]School of Artificial Intelligence and Data Science, University of Science and Technology of China
[2]Hong Kong University of Science and Technology (Guangzhou)
[3]Computer Network Information Center, Chinese Academy of Sciences
[4]Nanjing University of Aeronautics and Astronautics
wangchaoai@ustc.edu.cn, yixinsong@mail.ustc.edu.cn,
jye624@connect.hkust-gz.edu.cn, {chuanqin0426,dazh.shen}@gmail.com,
lfliu@mail.ustc.edu.cn, xiangwang1223@gmail.com, yanyongz@ustc.edu.cn

## Abstract

Recently, large language models (LLMs) have been explored for integration with collaborative filtering (CF)-based recommendation systems, which are crucial for personalizing user experiences. However, a key challenge is that LLMs struggle to interpret the latent, non-semantic embeddings produced by CF approaches, limiting recommendation effectiveness and further applications. To address this, we propose FACE, a general interpretable framework that maps CF embeddings into pre-trained LLM tokens. Specifically, we introduce a disentangled projection module to decompose CF embeddings into concept-specific vectors, followed by a quantized autoencoder to convert continuous embeddings into LLM tokens (descriptors). Then, we design a contrastive alignment objective to ensure that the tokens align with corresponding textual signals. Hence, the model-agnostic FACE framework achieves semantic alignment without fine-tuning LLMs and enhances recommendation performance by leveraging their pre-trained capabilities. Empirical results on three real-world recommendation datasets demonstrate performance improvements in benchmark models, with interpretability studies confirming the interpretability of the descriptors. Code is available in `https://github.com/YixinRoll/FACE`.

## 1 Introduction

Recommender systems are crucial in modern digital platforms, achieving personalized adaptation and driving user engagement across e-commerce, streaming services, and intelligent education [44, 39]. Collaborative filtering (CF) [8] methods, particularly those based on graph neural networks (GNNs), excel at capturing latent user-item relationships through embeddings in recommendation [38]. Recently, large language models (LLMs) have shown impressive reasoning capabilities in a wide range of tasks [21, 25, 24, 33, 45], prompting efforts to integrate them with recommendation systems. While some studies have explored enhancing recommendation quality with LLMs, a key challenge remains: LLMs, designed for processing natural language, are not inherently equipped to interpret the latent, non-semantic embeddings produced by CF methods. Bridging this gap, which allows LLMs to directly understand and utilize latent representations from CF models, remains an unresolved and pressing issue.

---

[†]Equal contribution.
[*]Corresponding author.

Existing approaches to integrating LLMs with recommender systems can be categorized into two types. The first directly feeds user/item textual information (e.g., title) into LLMs as static inputs [7, 19, 1]. As a representative example, TALLRec [2] performs recommendations using item titles and incorporates fine-tuning for recommendation tasks. However, without collaborative information, these methods often fall short of surpassing conventional recommenders. The second approach aligns the latent embedding space of CF with LLMs for understanding [53, 11, 48]. This is usually achieved through a unidirectional aligning network (e.g., MLP or Q-former [16]). For instance, ELM [31] equips LLMs with adapter layers for transforming abstract vectors into token embedding space; RLMRec [26] uses the contrastive or generative alignment of CF embeddings to inject textual knowledge into recommendation. While this spatial alignment helps, LLMs are pre-trained on natural textual language, and the mapped embeddings diverge from LLMs' original token space [51]. Consequently, the implicit alignment does not enable frozen LLMs to directly interpret the intrinsic meaning of CF embeddings. This limitation hampers LLMs' capacity to develop a deep understanding of user preferences and to support complex downstream recommendation tasks.

To enable universal compatibility between arbitrary CF models and LLMs, several technical challenges must be addressed: 1. *Decoupling Entangled Representations*: CF embeddings often combine multiple types of user preferences (or item features) into a single entangled vector [54], making it difficult to separate and interpret a user's preferences across different aspects. 2. *Continuous-to-Discrete Mapping*: Translating continuous embeddings into LLM-readable tokens without fine-tuning requires resolving the mismatch between continuous CF representations and discrete, high-dimensional pretrained LLM token embeddings. 3. *Semantic Consistency*: Mapped tokens must retain the users' semantic intent while aligning with LLMs' linguistic priors, ensuring that these representations can be effectively mapped to LLM token embeddings without distorting the information.

To address the above challenges, we propose FACE (a general Framework for mApping Collaborative filtering Embeddings into LLM tokens). Specifically, we first introduce a disentangled projection module that decomposes entangled CF embeddings into concept-specific vectors. We then design a quantized variational autoencoder [35] that learns a codebook to quantize continuous embeddings into discrete pretrained LLM textual tokens (named descriptors). Furthermore, we propose a contrastive alignment learning strategy that ensures the mapped tokens align with their corresponding textual signals. By achieving better semantic alignment, FACE not only establishes an accurate representation-semantic mapping between general CF models and LLMs without fine-tuning LLMs but also leverages the powerful capabilities of LLMs to enhance the performance of the original recommendation model and interpret its embedding. Finally, extensive experiments on three open-access datasets demonstrate that FACE improves the performance of conventional CF models, while interpretability studies validate that the decoded tokens align with textual signals and provide embedded interpretations for user-item interactions [12]. Our main contributions can be summarized as follows:

- To the best of our knowledge, this is the first work to propose a model-agnostic framework for directly mapping CF user/item embeddings into pre-trained LLM textual tokens, allowing LLMs to better understand user preferences and support more complex recommendation tasks.

- We introduce FACE, a general and interpretable representation alignment framework that leverages vector quantization and contrastive learning to efficiently and accurately convert entangled CF embeddings into pre-trained textual tokens.

- Extensive empirical results on three real-world datasets show that FACE enhances recommendation performance across various benchmark approaches, while interpretability studies demonstrate its great interpretability.

## 2 Related Work

**Collaborative Filtering.** Collaborative filtering (CF) is a technique used in recommendation systems that makes predictions based on preferences or similarities. Matrix factorization (MF) [14, 37] is the prototype of collaborative filtering based on representation learning, where the embedding matrix of entities (users & items) is learned directly. Inspired by the success of graph neural networks (GNNs) in modeling graph-structured data [13], many studies have explored graph-based representation learning by modeling the user-item bipartite graph structure. Simplified graph collaborative filtering models that eschew complex deep learning operations, such as LR-GCCF [3] and LightGCN [9], have demonstrated superior performance in experiments. Subsequently, numerous

studies have focused on optimizing graph collaborative filtering models. For instance, some have utilized self-supervised learning methods to enhance the robustness of representations [43, 50, 20], while others have addressed issues like over-smoothing [5] and over-correlation [46]. Despite these encouraging advances, pure collaborative filtering methods consistently struggle with data sparsity and explainability in practical applications.

**LLM-based Recommendations.**  Advances in LLMs have sparked significant research interests in enhancing recommender systems. Much prior work is devoted to utilizing or training LLMs to function as effective recommenders. A straightforward approach is to list the historical interaction item sequence and design prompts to guide LLMs to make recommendations [6, 7], and TALLRec [2] further incorporates tuning into LLMs for the recommendation task adaptation to bridge the gap between recommendation tasks and natural language tasks. Recently, research has shown that LLMs can comprehend recommendation embeddings via lightweight adapters [49, 31], and can be aligned with domain-specific latent embedding spaces [32]. Building on this, some works integrate collaborative information to improve LLM's performance in recommendations. For instance, CoLLM [53]/P4LM [11] combines mapped collaborative information with item titles into LLM recommendations for better performance/explanation; ILM [48] adopts the Q-former architecture to encode item CF embeddings for conversational recommendation. However, the mapped embeddings diverge from the LLM's original token space, which hinders the full utilization of the LLM's capabilities. To address this challenge, BinLLM [52] encodes collaborative information textually into an IP-address-like string. However, relying solely on numerical tokens limits the LLM's comprehension and still necessitates fine-tuning for LLMs to understand the collaborative information. Moreover, owing to the limited context window and high computational costs of LLMs, these LLM-based methods can hardly be applied to a full-ranking setting in real-world recommendation scenarios, thus suffering from scalability issues.

Another stream of work concentrates on enhancing existing collaborative filtering methods with the assistance of LLMs. Extracting and injecting the world knowledge and context comprehension abilities of LLMs into conventional recommender systems could enhance item and user modeling [30]. LLMRec [42] augments semantic and collaborative data with the power of LLMs, and RLMRec [26] aligns the final representation of users and items obtained from the backbone of a conventional recommender (e.g., LightGCN) with the profile generated from a text embedding model. However, the LLM's capabilities are not being fully utilized due to the large gap between text and recommendation. Differently, we encode the CF embeddings into the token space of the LLMs. By doing so, a pre-trained LLM can directly understand and interpret CF embeddings through its inherent literal knowledge, without requiring additional fine-tuning.

## 3 Methodology

### 3.1 Framework Overview

The FACE framework transforms CF embeddings into textual tokens (descriptors) compatible with LLMs. As illustrated in Figure 1, the framework consists of two main steps. In the first step, vector-quantized disentangled representation mapping (Section 3.2), we employ an AutoEncoder architecture with a quantized codebook of token embeddings following dimensionality reduction. The encoder, composed of a multi-projector and transformer, disentangles the original CF embeddings and learns complex relationships. In the second step, contrastive learning for semantic representation alignment (Section 3.3), we utilize an LLM embedding model to encode user/item summaries and descriptor-generated sentences, which are then aligned through a contrastive learning objective.

### 3.2 Vector-quantized Disentangled Representation Mapping

To enable effective mapping of CF embeddings with pre-trained LLM tokens, inspired by VQ-VAE [35], we propose to apply a frozen LLM vocabulary as the codebook, with the encoder-decoder structure to learn the mapping relationships.

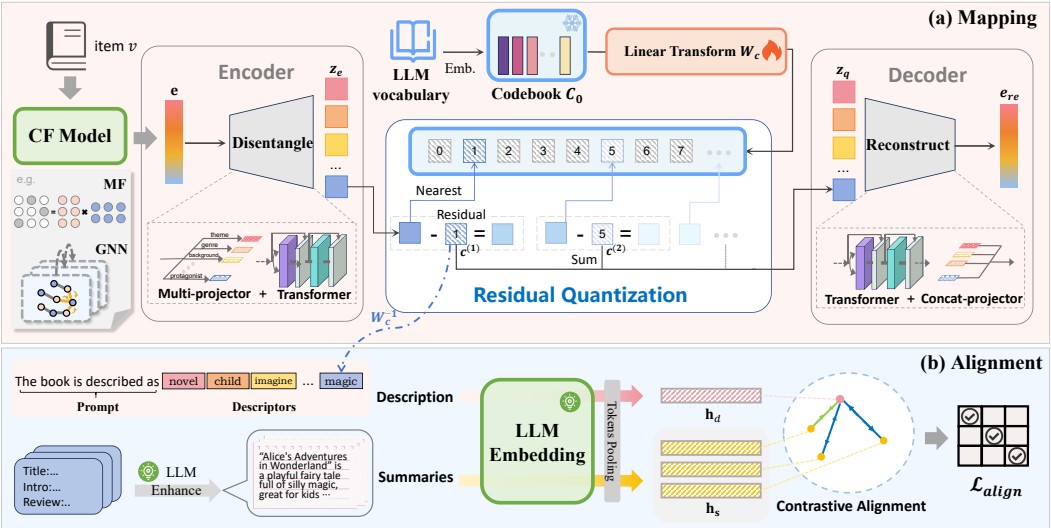

Figure 1: The overall architecture of our FACE framework. (a) Mapping stage. We employ a framework similar to the RQ-VAE architecture for the final embedding of a CF model with a frozen LLM codebook, encoding the CF embedding into pre-trained LLM tokens. (b) Alignment stage. We leverage contrastive learning to achieve the semantic alignment of descriptors and summaries.

### 3.2.1 Codebook for Quantization

This part focuses on constructing a codebook with LLM's vocabulary for quantization. First, we conduct filtering on the vocabulary of the LLM. Specifically, with the assistance of the Corpus of Contemporary American English (COCA) [4], we select words (no subwords) that carry meaningful semantic information for interpretation. Then, we obtain a subset $\mathcal{D} \subseteq \mathcal{D}_{\text{LLM}}$ of the LLM vocabulary, where $\mathcal{D}_{\text{LLM}}$ represents the full vocabulary of the language model. We freeze the embedding of these tokens as the codebook in our quantized autoencoder structure, formulated as:

$$C_0 = E_{\text{LLM}}(\mathcal{D}), \tag{1}$$

where $E_{\text{LLM}}$ denotes the pretrained token embedding component of the LLM, and the embedding matrix $C_0 \in \mathbb{R}^{|\mathcal{D}| \times d_{\text{LLM}}}$. These token embeddings encode rich semantic information, enabling further mathematical operations in the vector space to quantify semantic associations. To reduce redundancy in high-dimensional LLM token embeddings after filtering, we apply a trainable linear transformation:

$$C = W_c C_0, \tag{2}$$

where $W_c \in \mathbb{R}^{d \times d_{\text{LLM}}}$, and $d$ ($d < d_{\text{LLM}}$) is the dimension for quantization. This dimensionality reduction not only enhances computational efficiency but also initiatively aligns LLM tokens with CF embeddings. Crucially, we freeze the original codebook embeddings $C_0$ while exclusively updating the projection basis vectors through $W_c$. This design aligns with the SimVQ approach [55], where theoretical analysis demonstrates that optimizing the latent space geometry rather than individually selected code vectors mitigates representation collapse [29].

### 3.2.2 Representation Disentanglement and Quantization

To ensure the integrity of information as far as possible while mapping the CF representation to the vocabulary space, we adopt the architecture of the autoencoder, which is designed to learn a latent space that is capable of recovering the original data. The encoder is a disentangled projection module consisting of two main parts.

First, the final representation $e$ of the backbone CF model passes through a multi-projector, which projects this vector into $n$ different directions, capturing information from various perspectives:

$$e_i = W_i e \quad \text{for} \quad i = 1, 2, \ldots, n, \tag{3}$$

where $W_i$ denotes the weight matrix of the $i$-th projection head. The projection heads are initialized orthogonally. These weight matrices project the origin embedding into $n$ new spaces, aiming to disentangle the complex preferences implied in the CF embedding into multiple aspects.

Secondly, the obtained $n$ vectors are encoded using a transformer encoder module [36] to capture the relationships between them and perform non-linear mapping:

$$(z_{e_1}, z_{e_2}, ..., z_{e_n}) = \text{Transformer}_e(e_1, e_2, ..., e_n). \tag{4}$$

**Residual Quantization.** For each $z_e$, it is then quantized with the codebook $C = (c_1, c_2, \ldots c_{|\mathcal{D}|})$ constructed in Section 3.2.1 by recurrently selecting the nearest codeword to the current residual measuring by Euclidean distance. For layer $h$, the residual quantization (RQ) [15] is expressed as:

$$r^{(h+1)} = r^{(h)} - c_k^{(h)} \quad \text{w.r.t.} \quad k = \arg\min_j \|r^{(h)} - c_j\|_2^2. \tag{5}$$

where $r_h$ is the residual vector in the $h$-th RQ level, and initially $r^{(1)} = z_e$. After H levels of residual quantization, the quantization embedding of $z_e$ can be obtained by $z_q = \sum_{i=1}^{H} c^{(h)}$. To facilitate backpropagation through this non-differentiable operation, the Straight-Through Estimator (STE) is applied between $z_e$ and $z_q$. Compared with vanilla quantized autoencoder (e.g., VQ-VAE), RQ quantizes residuals through multiple stages to achieve more precise discrete representations and higher reconstruction quality. The hierarchical structure of residual quantization allows the model to capture different levels of semantic information. Unlike conventional residual quantization approaches, our implementation shares the same codebook across all quantization levels, with this codebook being composed of pre-trained LLM token embeddings, maintaining semantic consistency among the quantized tokens at different hierarchical levels. When obtaining $z_q$ by progressively summing the quantized vectors from each layer, it can be implicitly interpreted as a composition of meaningful semantic tokens. The descriptors can be obtained from the first-level quantization $c^{(1)}$, as the nearest vector captures the main information of $z_e$. For simplicity, we denote the descriptor embedding as $z_{d_i} = c^{(1)}$ in aspect $i$.

Similarly, the decoder consists of a transformer followed by a concat-projector to fuse the disentangled quantized embedding to get the recovered CF embedding $e_{re}$. The transformer in the decoder shares the reverse architecture of that in the encoder. And it seems like a simplistic embedding model, which generates the embedding for recommendations from several words. Furthermore, the decoder can be used as a generator: given a few keywords, the well-trained decoder can generate a primary embedding for users/items. The decoder can be expressed as:

$$(p_1, p_2, ..., p_n) = \text{Transformer}_d(z_{q_1}, z_{q_2}, ..., z_{q_n}), \tag{6}$$

$$e_{re} = W \, \text{Concat}(p_1, p_2, \ldots, p_n) + b. \tag{7}$$

The loss function for mapping the embedding into descriptors appears similar to the RQ-VAE loss. The reconstruction loss ensures that the framework can effectively reconstruct the input embedding $e$, while the quantization loss constrains the encoder and the codebook, ensuring the encoder output $z_e$ is close to the codebook vector $z_q$ and vice versa. It can be formulated as (sg[·] denotes stop gradient):

$$\mathcal{L}_{recons} = \log p(e|z_q) = \|e_{re} - e\|_2^2, \tag{8}$$

$$\mathcal{L}_Q = \sum_{h=1}^{H} (\|\text{sg}[r^{(h)}] - c_k^{(h)}\|_2^2 + \beta \|\text{sg}[c_k^{(h)}] - r^{(h)}\|_2^2), \tag{9}$$

$$\mathcal{L}_{map} = \mathcal{L}_{recons} + \mathcal{L}_Q. \tag{10}$$

### 3.3 Contrastive Learning for Semantic Representation Alignment

To achieve semantic alignment between CF representations and LLM-derived embeddings and ensure the interpretability of mapped descriptors, we propose a contrastive representation alignment strategy, the goal of which is to align the synthetic semantics of descriptors with the corresponding textual information of users/items, ensuring the coherence of the mapped descriptors with the text.

### 3.3.1 Semantic Embedding with LLM

Due to the excellent comprehensive capability of LLM, high-quality text embeddings can be obtained from LLM, especially those adapted for embedding tasks [22], through token pooling. We will show how to process text from summaries and descriptors and generate semantic embeddings from LLMs.

**Summary Embedding:** The summary embeddings are derived from textual information in the dataset, and they serve as the anchor for alignment. Following the paradigm established in RLMRec [26], we first generate a structured textual summary, denoted as $\mathbf{s}$, for each user and item. This is achieved by employing an LLM to enhance the original raw text. The process can be formulated as:

$$\mathbf{s} = \text{LLM}(\mathbf{P}_s \oplus \mathbf{T}_{ori}), \tag{11}$$

where $\mathbf{P}_s$ is a predefined prompt and $\oplus$ denotes the concatenation operation. The original text, $\mathbf{T}_{ori}$, is compiled from different sources: for items, it consists of the title, description, attributes, and any available user reviews; for users, it is composed of the summaries of their historically interacted items. Once the enhanced summary $\mathbf{s}$ is generated, it is encoded into a dense vector representation $\mathbf{h}_s$ using an LLM-based embedding model $\mathcal{E}$:

$$\mathbf{h}_s = \mathcal{E}(\mathbf{s}). \tag{12}$$

**Descriptors Embedding:** The embedding process of descriptors is designed to generate comprehensive semantic representations for users and items from their extracted descriptors. The first step is to format the descriptors into a sentence-like input. We prepend a prompt $\mathbf{P}_d$ to the descriptors, which specifies the entity type (user or item) and indicates that the subsequent descriptors are keywords. For example, in a book recommendation context, the prompt for a user could be: "The reader and his preference can be described as:". However, directly generating embeddings from the textual form of the descriptors is a non-differentiable operation, which prevents gradient flow during training. To ensure end-to-end differentiability, we map the descriptors $(z_{d_1}, z_{d_2}, \ldots, z_{d_n})$ back to the original high-dimensional word embedding space. This is accomplished by applying an inverse transformation via the pseudo-inverse matrix $W_c^{-1} = (W_c^T W_c)^{-1} W_c^T$. These recovered embeddings are then concatenated with the prompt's token embeddings to form the final input sequence $\mathbf{d}$ as a *description* of the user/item:

$$\mathbf{d} = E_{\text{LLM}}(\mathbf{P}_d) \oplus \left( W_c^{-1} z_{d_1}, W_c^{-1} z_{d_2}, \ldots, W_c^{-1} z_{d_n} \right). \tag{13}$$

Finally, the complete sequence of embeddings $d$ is passed directly to the LLM encoder $\mathcal{E}$ to produce the final descriptors embedding $\mathbf{h}_d$:

$$\mathbf{h}_d = \mathcal{E}(\mathbf{d}). \tag{14}$$

### 3.3.2 Contrastive Alignment Learning Strategy

To train the words and align the semantics, we adopt the contrastive alignment learning loss [23] $\mathcal{L}_{align}$ to learn the distinctive features of users and items, which can be represented as follows:

$$\mathcal{L}_{align} = -\frac{1}{|\Omega|} \sum_{v \in \Omega} \log \frac{\phi(d_v, s_v)}{\sum_{v' \in \Omega} \phi(d_v, s_{v'})}, \tag{15}$$

where $\Omega$ is the current batch of data containing users and items, and $\phi(d, s)$ is a function that computes the matching score between descriptor-generated sentence $d$ and summary $s$. In this paper, we adopt the cosine similarity with temperature $\tau$:

$$\phi(d, s) = \exp(\frac{1}{\tau} \cos(\mathbf{h}_d, \mathbf{h}_s)). \tag{16}$$

This contrastive alignment learning strategy minimizes the distance between descriptors in $d_v$ and their corresponding fixed summary $s_v$, while pushing $d_v$ away from non-corresponding summaries $s_{v'}$. By contrasting positive $(d_v, s_v)$ pairs against negatives $(d_v, s_{v'})$, the descriptors adapt to emphasize unique attributes of their paired user/item, rather than generic features. $\mathbf{h}_s$ can be generated in advance and fixed during training, and it serves as stable semantic anchors, ensuring learned descriptors capture discriminative features for alignment.

Table 1: Overall performance comparison on Amazon-book, Yelp, and Steam datasets.

| Dataset | Amazon-book | | | | Yelp | | | | Steam | | | |
|---|---|---|---|---|---|---|---|---|---|---|---|---|
| | R@5 | R@20 | N@5 | N@20 | R@5 | R@20 | N@5 | N@20 | R@5 | R@20 | N@5 | N@20 |
| GMF | 0.0615 | 0.1531 | 0.0616 | 0.0922 | 0.0372 | 0.1052 | 0.0433 | 0.0660 | 0.0523 | 0.1343 | 0.0567 | 0.0844 |
| + FACE | 0.0658 | 0.1553 | 0.0659 | 0.0955 | 0.0414 | 0.1120 | 0.0483 | 0.0717 | 0.0547 | 0.1411 | 0.0594 | 0.0888 |
| LightGCN | 0.0659 | 0.1563 | 0.0657 | 0.0961 | 0.0421 | 0.1141 | 0.0488 | 0.0726 | 0.0530 | 0.1361 | 0.0584 | 0.0862 |
| + FACE | 0.0705 | 0.1622 | 0.0705 | 0.1009 | 0.0446 | 0.1203 | 0.0519 | 0.0766 | 0.0559 | 0.1439 | 0.0611 | 0.0912 |
| SimGCL | 0.0695 | 0.1617 | 0.0693 | 0.1001 | 0.0447 | 0.1209 | 0.0529 | 0.0775 | 0.0550 | 0.1420 | 0.0605 | 0.0899 |
| + FACE | 0.0747 | 0.1670 | 0.0737 | 0.1047 | 0.0461 | 0.1225 | 0.0534 | 0.0781 | 0.0594 | 0.1487 | 0.0649 | 0.0951 |
| LightGCL | 0.0810 | 0.1712 | 0.0816 | 0.1114 | 0.0452 | 0.1228 | 0.0530 | 0.0780 | 0.0526 | 0.1234 | 0.0576 | 0.0815 |
| + FACE | 0.0832 | 0.1759 | 0.0842 | 0.1148 | 0.0455 | 0.1253 | 0.0533 | 0.0793 | 0.0528 | 0.1238 | 0.0585 | 0.0818 |
| RLMRec | 0.0669 | 0.1572 | 0.0663 | 0.0981 | 0.0426 | 0.1165 | 0.0495 | 0.0737 | 0.0545 | 0.1408 | 0.0599 | 0.0887 |
| + FACE | 0.0679 | 0.1581 | 0.0672 | 0.0985 | 0.0435 | 0.1196 | 0.0503 | 0.0755 | 0.0556 | 0.1432 | 0.0604 | 0.0901 |

## 3.4 Optimization

The above framework is employed on the final representation $e$ of the basic collaborative filtering model $\mathcal{R}$, where prediction is performed on this representation. In other words, our proposed approach is model-agnostic. Any model that can perform representation learning for users/items can be aligned with LLMs through the framework. Assuming the optimization function of the recommender $\mathcal{R}$ is denoted as $\mathcal{L}_{\mathcal{R}}$, the overall optimization objective $\mathcal{L}$ can be formulated with coefficient $\mu$ and $\lambda$ :

$$\mathcal{L} = \mathcal{L}_{\mathcal{R}} + \mu\mathcal{L}_{map} + \lambda\mathcal{L}_{align}. \tag{17}$$

Nevertheless, directly optimizing the joint objective $\mathcal{L}$ would lead to unstable training dynamics. This is because simultaneously learning the recommendation backbone parameters from scratch and the discrete LLM mapping function creates conflict. For the sake of training stability, we adopt a 3-step training strategy. In step 1, pre-train the backbone recommender independently of our framework; in step 2, employ the quantized autoencoder structure without alignment to primarily map CF embeddings into LLM token embedding space; in step 3, add the semantic alignment to the framework and jointly optimize with the whole objective $\min \mathcal{L}$. This curriculum paradigm ultimately enables the alignment from the recommender to the LLM while enhancing its original capability and interpretability.

## 4 Experiments

In this section, we conduct extensive experiments to evaluate the performance of FACE in comparison to various state-of-the-art models across three real-world datasets. Specifically, we validate the performance of the recommendation, the interpretability of mapped descriptors, the ablation studies, and the sensitivity of the model.

### 4.1 Baselines

We evaluate the effectiveness of our framework by integrating it with five widely used state-of-the-art recommendation models: GMF [14]: decomposes the interaction matrix into latent representation; LightGCN [9]: a simplified graph convolutional network removes conventional neural components; SimGCL [50]: a contrastive learning framework without explicit data augmentation, generating multiple views via perturbations of the embedding; LightGCL [28]: leverages SVD to generate self-augmented representations; RLMRec [26]: aligns existing CF models with LLM by maximizing mutual information. Its contrastive variant based on LightGCN is used as the base model.

### 4.2 Experimental Settings

**Evaluations:** The evaluation protocols for recommendations employ two widely adopted ranking metrics: Recall@N and normalized discounted cumulative gain (NDCG@N) [10] with N={5, 20}, calculated through an all-ranking evaluation strategy [40]. These metrics consider all non-interacted items as potential candidates for each user during recommendation.

**Datasets:** We conduct experiments of the base model and our FACE framework on three public datasets: Amazon-book, Yelp, and Steam. Please refer to Appendix A.1 for dataset details.

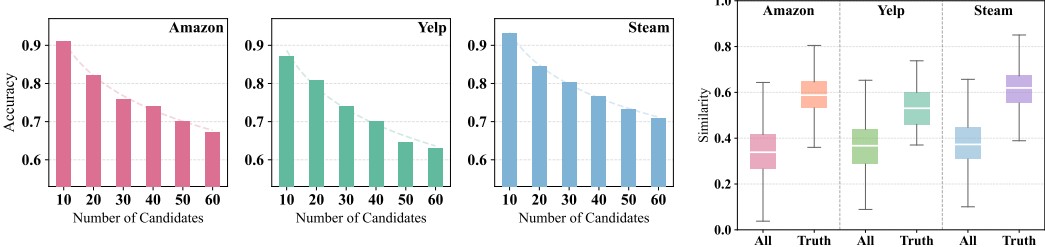

Figure 2: Item-retrieval task.

Figure 3: Item-generation task.

**Implementation Details:** Our proposed framework is a model-agnostic plugin. Therefore, the recommendation objective is that of the base model. In our experiments, the collaborative filtering base models all adopted the BPR recommendation loss function as their objective. To ensure fair comparisons, the latent embeddings for all baselines and FACE are set to 256. We determine the main hyperparameters through a grid search. For FACE, we use the pre-trained embedding model adapted from LLaMA2-7B[34] as the LLM. Additionally, all three datasets are divided into training, validation, and test sets with a split of 3:1:1. FACE is trained using the Adam optimizer, with a learning rate 1e-2 and batch size 128. We implement the model with SSLRec[27] and deploy it on NVIDIA A100.

### 4.3 Performance Comparisons

To validate FACE's effectiveness in enhancing the performance of recommender backbones, we conduct a comparative analysis of several base models integrated with FACE against state-of-the-art methods in collaborative filtering and the LLM-enhanced recommendation model. We run the models three times and report the average results, which leads us to the following conclusions: (1) FACE enhances the performance of various backbones, with maximum gains of 7.31%, 11.55%, and 8.00% on the Amazon-book, Yelp, and Steam datasets, and nearly all sorts of CF models can benefit from FACE's integration. The empirical results highlight FACE framework can be used as a general descriptor generator without the loss of performance and could even serve to enhance the existing CF model from the text alignment. (2) Moreover, our framework can be applied to RLMRec, a collaborative filtering model that already incorporates textual information, to achieve further performance improvement, even if the improvement is minor compared to its application on traditional recommendation models. These comparative results demonstrate that our proposed framework, especially its descriptor-based alignment strategy with large language models, exhibits superior effectiveness in enhancing CF performance.

### 4.4 Interpretability Studies

#### 4.4.1 Item Recovery Based on Descriptors

**Item-retrieval Task:** To demonstrate that the implicit semantic information in descriptors can be understood by LLMs. We first introduce an item-retrieval task, aiming to leverage the linguistic capacity of LLM to retrieve the original item based solely on the FACE-generated descriptors. The candidates are $l$ items, each of which is represented by its basic information (e.g. title and summary). We ask the LLM to select the most relevant item from the candidates based on the descriptors, and the retrieval process is evaluated by the accuracy, as shown in Figure 2. The result primarily indicates that the descriptors can effectively capture the semantic information of the items, as the LLM is able to retrieve the original item with a high probability. However, the accuracy of the item-retrieval task is affected by the number of candidates $l$, and the more candidates there are, the more challenging it becomes for the LLM to retrieve the original item.

**Item-generation Task**: Going a step further, we propose an additional item-generation task in which the LLM is asked to generate both an item and its description based solely on the given descriptors. To assess the quality of the generated items, we follow the methodology outlined in Section 3.3.1 to obtain their summary embeddings. Then, we calculate the cosine similarity between the summary

embeddings of the generated items and those of the original items in the dataset. Ideally, the generated items should be similar to the truth item (the source of these descriptors). We report the distribution of cosine similarity between generated items and all the original items (All), and compare it with that between generated items and their corresponding truth items (Truth) in Figure 3. The whiskers represent the non-outlier range, with outliers defined as values beyond 1.5×IQR from Q1 or Q3.

The results show that the generated item is significantly more similar to the truth item than to others in the dataset, indicating that the descriptors can effectively capture the semantic information of the items. This suggests that LLMs can understand and generate items based on the descriptors, demonstrating the quality of FACE-generated descriptors and their potential for enhancing the interpretability of recommendation systems.

### 4.4.2 Real User Study on Interaction Interpretation

We conduct a real-user evaluation to assess how convincingly descriptor-based interpretations reflect the implicit relevance of user-item interactions, focusing specifically on the interpretability of descriptors in explaining user-item interactions, and the key lies in whether the LLM can understand the implicit relevance of these descriptors. We ask LLM to generate interpretations for user-item interactions based on the descriptors, and then we ask human annotators and another LLM to rank the interpretations based on their reliability. This experiment is designed to evaluate the quality of the generated explanations and to assess how well LLM can understand and explain.

To be specific, we conduct the interaction interpretation experiment, respectively based on RLMRec profiles (LLM-generated profiles from textual information) and FACE descriptors. We randomly sample 40 users from the Amazon dataset along with their interacted items. For each user, one interacted item from the test set is selected as a positive sample, while three non-interacted items are chosen as negative samples. Next, we provide LLM with the user and item profiles/descriptors and instruct it to assume an interaction between them, prompting it to generate an explanation grounded in the implicit relevance of the given information. Finally, we ask ten human annotators and another LLM (DeepSeek v3) to rank the four candidate explanations for each user based on their reliability, where higher-ranked explanations are expected to correspond to the positive samples.

The results in Table 2 indicate that the relevance between sets of descriptors can be utilized by LLMs to interpret interactions. Furthermore, the ranking results show that the explanations generated based on FACE descriptors are slightly more reliable than those based on RLMRec profiles, and it is worth noting that a set of descriptors contains only 16 tokens compared to a paragraph of profile, suggesting that descriptors can be more efficient in capturing the implicit relevance of semantic information in user-

Table 2: Ranking Results.

| Method | Manual | LLM |
|---|---|---|
| RLMRec Profile | 1.935 | 1.800 |
| FACE Descriptors | 1.915 | **1.700** |

item interactions and providing explanations. These interpretability studies quantify the quality of semantic mapping and suggest that our method improves the interpretability of CF embedding.

### 4.5 Hyperparameter Analysis

In this part, we carry out a hyperparameter analysis on the Amazon dataset with GMF+FACE and LightGCN+FACE, concentrating on three crucial hyperparameters: descriptor number $n$, codebook dimension $d$, and the alignment weight $\lambda$. The results can be observed in Figure 4.

**Codebook Dimension** means the degree of semantic preservation for tokens before and after dimensionality reduction. Lower dimensions (64D) limit the semantic capacity of tokens, whereas higher dimensions (512D) may cause overfitting.

**Descriptor Number** plays a significant role in reconstruction and alignment, and meanwhile, it means the number of components after disentanglement. Increasing descriptors from 1 to 16 demonstrates an overall upward trend in performance, indicating that more components and descriptors preserve the preference contained in CF embeddings more comprehensively. However, 8 to 16 descriptors already possess sufficient information. When it is larger, descriptors suffer from duplication.

**Alignment Weight** controls the injection of textual signals and significantly influences the performance. A larger $\lambda$ injects more text information into the CF model. However, when it is too large,

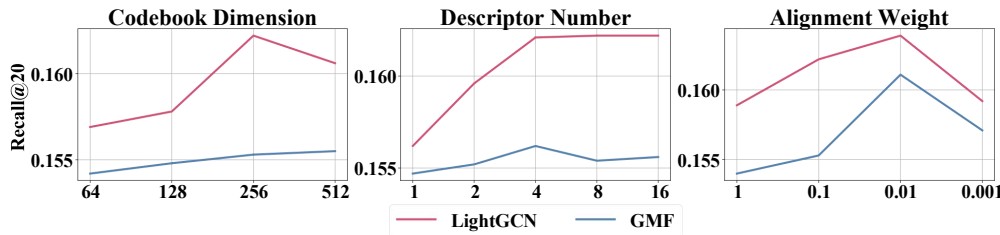

Figure 4: Analysis results on hyperparameter sensitivities.

the recommendation task will be hindered. Conversely, when it is relatively low, the performance drops with a poor alignment result.

## 4.6 Ablation Studies

We conduct ablation studies on Amazon-book and Yelp datasets with LightGCN as the base model to validate the necessity of key components in our framework. Four variants are compared: (1) Full: the complete FACE model; (2) w/o trans: FACE with the transformer module removed from the autoencoder; (3) w/o recons: FACE excluding the decoder for reconstructing the original input embeddings; (4) w/o align: FACE without the semantic representation alignment step. The results in Table 3 show that the full model achieves the best performance on both datasets.

Although the transformer module has a relatively minor overall impact, its complex transformation process can enhance the quality of the descriptors, as the self-attention mechanism enables effective communication among disentangled embeddings. Additionally, the absence of reconstruction negatively impacts the performance, and it is noted that in this variant, descriptors suffer from the lack of diversity, resulting in the loss of information in the one-way mapping process. Notably, without alignment, the metric drops dramatically, demonstrating

Table 3: Ablation studies.

| Dataset | Variant | Recall@20 | NDCG@20 |
|---|---|---|---|
| Amazon-book | Full | 0.1622 | 0.1009 |
| | w/o trans | 0.1611 | 0.0994 |
| | w/o recons | 0.1586 | 0.0981 |
| | w/o align | 0.1565 | 0.0962 |
| Yelp | Full | 0.1203 | 0.0766 |
| | w/o trans | 0.1200 | 0.0762 |
| | w/o recons | 0.1191 | 0.0760 |
| | w/o align | 0.1171 | 0.0741 |

that the step of contrastive learning for semantic representation alignment plays a significant role in enhancing performance by equipping CF models with semantic information.

## 5 Conclusion

In this paper, we introduced FACE, a novel framework that bridges the gap between collaborative filtering models and large language models by mapping CF embeddings into the semantic tokens of LLMs. Through a disentangled projection module and a vector-quantized variational autoencoder, FACE efficiently converts continuous CF embeddings into discrete, LLM-compatible tokens. A contrastive alignment objective ensures that these tokens maintain semantic consistency, enhancing the interpretability and performance of recommendation systems. Our extensive experiments on three real-world datasets demonstrate that FACE can improve the performance of various CF models. Additionally, interpretability studies confirm the improved interpretability of the descriptors.

## Acknowledgments and Disclosure of Funding

This work was supported in part by the National Natural Science Foundation of China (Grant No. 62506348), the Natural Science Foundation of Anhui Province (Grant No. 2508085QF211), the CCF-1688 Yuanbao Cooperation Fund (Grant No. CCF-Alibaba2025005), the National Natural Science Foundation of China (No. 62406141), the China Postdoctoral Science Foundation (No. GZC20252740), the National Natural Science Foundation of China (Grant No. 62332016), and the iFLYTEK Cooperation Project.

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

# A   More Implementation Details

## A.1   Dataset Details

We conduct experiments on three benchmark datasets: Amazon-book, Yelp and Steam: **Amazon-book**[2], derived from the Amazon Product Review corpus, focuses on literary products, containing user-book interactions, review texts, and product descriptions; **Yelp**[3], which consists of user-business interactions from metropolitan services, featuring categorical attributes and consumer feedback; and **Steam**[4], collected from the digital game distribution platform, containing user-game engagements with gameplay reviews. Dataset statistics are shown in Table 4.

Table 4: Statistics of the datasets.

| Dataset | Users # | Items # | Interactions # | Density |
|---|---|---|---|---|
| Amazon-book | 11,000 | 9,332 | 120,464 | $1.2e^{-3}$ |
| Yelp | 11,091 | 11,010 | 166,620 | $1.4e^{-3}$ |
| Steam | 23,310 | 5,237 | 316,190 | $2.6e^{-3}$ |

## A.2   Hyperparameter Settings

For our FACE framework, the number of descriptors is set to 16, and the dimension of the codebook is set to 256. The temperature of alignment loss is fixed to 0.02, and the coefficients $\beta$, $\mu$ and $\lambda$ are set to 0.25, 1 and 0.1, respectively. For the parameters of baselines or base models, we mainly use the official setting from the SSLRec benchmark platform for fair comparisons, except for those we have stressed in the paper.

# B   Performance with Other LLMs

We implement our FACE framework using several more representative LLMs, including all-MiniLM-L6-v2 [41] and gte-Qwen2-7B-instruct [18], to evaluate its generality across different language model architectures and scales. Table 5, 6, 7 reports the performance of FACE when integrated with these LLMs on the Amazon-book, Yelp, and Steam datasets. The metric I@N denotes the accuracy of the item-retrieval task (Section 4.4.1) with N candidates.

Table 5: Performance of FACE with different LLMs on the Amazon-book dataset.

| Model | R@5 | R@20 | N@5 | N@20 | I@10 | I@50 |
|---|---|---|---|---|---|---|
| LightGCN | 0.0659 | 0.1563 | 0.0657 | 0.0961 | - | - |
| + FACE (LLaMA2) | 0.0705 | 0.1622 | 0.0705 | 0.1009 | 0.9116 | 0.7024 |
| + FACE (MiniLM) | 0.0703 | 0.1631 | 0.0698 | 0.1012 | 0.9105 | 0.7394 |
| + FACE (Qwen2) | 0.0692 | 0.1623 | 0.0682 | 0.1006 | 0.8703 | 0.6291 |

Table 6: Performance of FACE with different LLMs on the Yelp dataset.

| Model | R@5 | R@20 | N@5 | N@20 | I@10 | I@50 |
|---|---|---|---|---|---|---|
| LightGCN | 0.0421 | 0.1141 | 0.0488 | 0.0726 | - | - |
| + FACE (LLaMA2) | 0.0446 | 0.1203 | 0.0519 | 0.0766 | 0.8712 | 0.6469 |
| + FACE (MiniLM) | 0.0445 | 0.1191 | 0.0518 | 0.0762 | 0.8907 | 0.6322 |
| + FACE (Qwen2) | 0.0449 | 0.1193 | 0.0517 | 0.0761 | 0.8356 | 0.6158 |

---

[2]http://jmcauley.ucsd.edu/data/amazon/

[3]https://business.yelp.com/data/resources/open-dataset/

[4]https://www.kaggle.com/datasets/tamber/steam-video-games/data

Table 7: Performance of FACE with different LLMs on the Steam dataset.

| Model | R@5 | R@20 | N@5 | N@20 | I@10 | I@50 |
|---|---|---|---|---|---|---|
| LightGCN | 0.0530 | 0.1361 | 0.0584 | 0.0862 | - | - |
| + FACE (LLaMA2) | 0.0559 | 0.1439 | 0.0611 | 0.0912 | 0.9330 | 0.7322 |
| + FACE (MiniLM) | 0.0557 | 0.1425 | 0.0608 | 0.0899 | 0.9287 | 0.7079 |
| + FACE (Qwen2) | 0.0558 | 0.1436 | 0.0611 | 0.0908 | 0.8753 | 0.6194 |

The results demonstrate that FACE consistently enhances interpretability while maintaining competitive recommendation accuracy across different underlying LLMs, underscoring the robustness and flexibility of our framework. Notably, the three LLMs achieve comparable performance when integrated with FACE, indicating that the choice of LLM (e.g., smaller models like MiniLM or larger ones like Qwen2) does not substantially affect the framework's effectiveness. However, Qwen2 exhibits marginally inferior performance, likely attributable to its specialization in query-document matching rather than semantic similarity. These findings reinforce that FACE is model-agnostic and can be seamlessly adapted to diverse LLMs, rendering it suitable for a broad spectrum of practical recommendation scenarios.

Furthermore, this approach reveals the potential to employ small-scale language models for vocabulary and embedding generation, thereby significantly improving efficiency and reducing computational resource demands when mapping CF embeddings to semantic tokens. Subsequently, large language models with superior generative capabilities can be leveraged for downstream tasks—such as explanation generation, large-model-based recommendations, and controllable recommendations—building upon the descriptors produced by the smaller-model-based FACE.

## C  Performance Comparing with Other LLM4Rec Methods

To fairly compare the ability of other frameworks to align CF with LLMs for textual information injection and performance improvement, we adapt CTRL [17] and KAR [47] on the base model LightGCN, and investigate the effect of further applying FACE to these LLM-enhanced recommendation models. The experimental results in Table 8 demonstrate that FACE can be effectively applied to various existing LLM-enhanced recommendation approaches to further improve their performance. These findings are consistent with those presented in the original paper, highlighting the advantages of the FACE framework in utilizing semantic mapping to enable LLMs to comprehend collaborative embeddings.

Table 8: Performance comparison of FACE-enhanced models on three datasets. The best results are highlighted in bold.

| Dataset | Metric | CTRL | CTRL+FACE | KAR | KAR+FACE |
|---|---|---|---|---|---|
| **Amazon** | R@5 | 0.0685 | 0.0699 | 0.0671 | **0.0712** |
| | R@20 | 0.1566 | **0.1612** | 0.1547 | 0.1559 |
| | N@5 | 0.0666 | 0.0706 | 0.0660 | **0.0714** |
| | N@20 | 0.0963 | **0.1013** | 0.0952 | 0.1002 |
| **Yelp** | R@5 | 0.0413 | 0.0432 | 0.0408 | **0.0446** |
| | R@20 | 0.1161 | 0.1189 | 0.1132 | **0.1249** |
| | N@5 | 0.0471 | 0.0510 | 0.0476 | **0.0518** |
| | N@20 | 0.0719 | 0.0755 | 0.0712 | **0.0781** |
| **Steam** | R@5 | 0.0531 | **0.0569** | 0.0502 | 0.0516 |
| | R@20 | 0.1375 | **0.1461** | 0.1322 | 0.1349 |
| | N@5 | 0.0586 | **0.0620** | 0.0553 | 0.0571 |
| | N@20 | 0.0865 | **0.0922** | 0.0826 | 0.0852 |

# D  Case Studies

## D.1  Example for Item-Generation Task

To verify the quality of the descriptors mapped by the model, Section 4.4.1 proposes an item-generation task in which the LLM is asked to generate both an item and its description based solely on the given descriptors (prompt template in Appendix E). And then we generate its summary to compare it with the original one. To facilitate a deeper understanding of the process, we present a case study as illustrated in Figure 5. It can be discovered that the summary of the generated item closely resembles the original summary (the summary of the corresponding item in the dataset), and is not similar to the non-corresponding item. This indicates that the descriptors generated by our FACE framework are effective in capturing the semantic information.

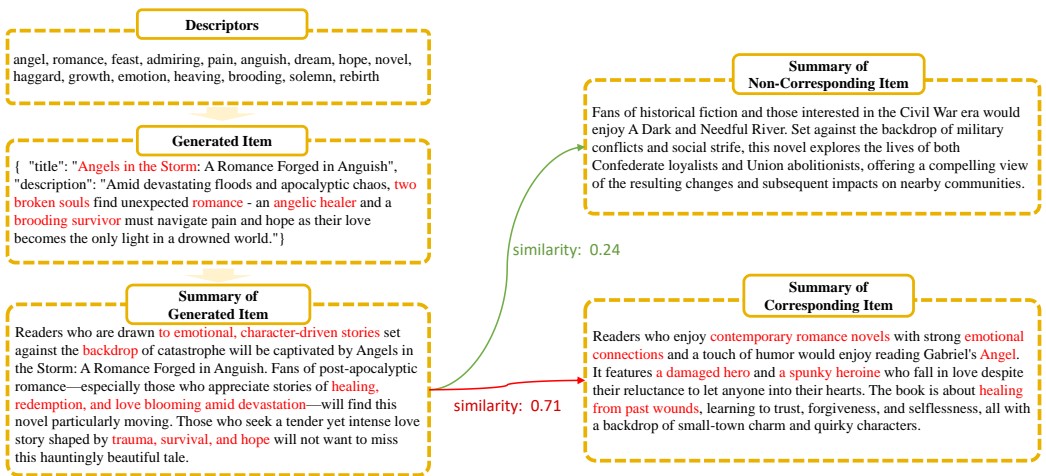

Figure 5: An example for item generation task.

## D.2  Example for Interaction Interpretation

In Section 4.4.2, we perform a real user study on interaction interpretation, which guides LLM to provide an explanation on why the user would like the item. We present a case study in Figure 6 to simply illustrate the process. The descriptors of the user and items (including positive items and negative items in the test set) are generated by our FACE framework, and the LLM is prompted to act as an interpretable recommender to provide an explanation for user-item interaction (prompt template in Appendix E). The generated explanations are then evaluated by real users and LLMs. Ideally, the explanations provided by LLM for actual interactions (a user and its positive item) should be more reliable and persuasive, achieving a high ranking.

The results show that the explanations of interactions between users and positive items are more credible than those between users and negative items. This demonstrates that the relevance between the descriptors of users and items can be effectively understood by LLMs to generate reasonable explanations for user-item interactions.

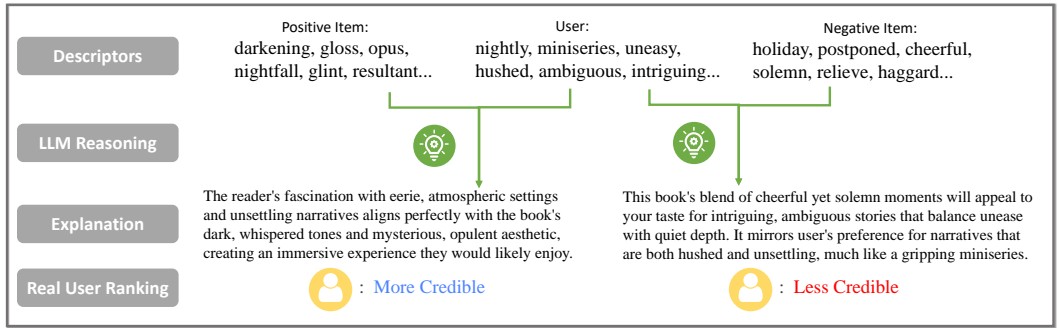

Figure 6: An example for interaction interpretation.

# E  Prompt Template

For summary generation, we use the same prompt template as in the original paper of RLMRec.

For the item-retrieval task, we use the following prompt template to search for the most relevant item from the list of candidates based on the descriptors:

---

**Prompt for *Item Retrieval Task***

**### Instruction**
Given a set of descriptors of an item, find the most relevant item from the list of candidates. The descriptors are a list of keywords that describe the item. The candidates are a list of items with their profiles.
Request:
1. Provide the id of the most relevant item from the list of candidates. For example, your reply 'item1'.
2. Do not provide any other information or explanation.
3. Try to find the association between the descriptors and the profiles.
**### Input**
The descriptors of the item are: *<descriptors>*. The candidates are: *<candidate list>*. Please provide the id of the most relevant item from the list of candidates.

---

For item generation task, we use the following prompt template to generate a new item based on the descriptors given by FACE:

---

**Prompt for *Item Generation* based on Descriptors**

**### Instruction**
Based on the item descriptors, generate a title and description of the item :
1. The provided item is from *<dataset name, e.g., Amazon>* dataset, where the item is a *<dataset item, e.g., book>*.
2. The descriptors are a list of keywords that describe the item.
3. The title is not required to be an actual title, but it should be a title-like phrase.
4. The description should be one or two sentences that introduce the item.
5. Please provide your answer in JSON format, following this structure: { 'title': '...', 'description': '...' }.
Examples:
*<examples>*.
**### Input**
The descriptors of the item are: *<descriptors>*

---

In real user study, we generate the explanation for the user-item interaction based on the FACE descriptor sets or RLMRec profiles of the user-item pair. The LLM is prompted to act as a recommender and provide an explanation for the interaction. Specifically, we randomly select 40 users from Amazon-book dataset. For each user, we randomly select 1 item from their interacted items in

the test set and 3 randomly selected items. In other words, we have 4 user-item pairs for each user: 1 positive pair and 3 negative pairs. We generate the explanations for these 4 pairs.

Based on FACE descriptors, the prompt template is as follows:

---

**Prompt for *Interaction Interpretation* based on Descriptors**

### Instruction
Task: Act as an explanation generator for recommendation systems. Given a pair of user(reader) and item(book) descriptor sets, you need to generate a short explanation of why the user would like the item.
Note:
1. Assume that the user will interact with (like) the item, whatever their descriptors are.
2. Provide your evidence to support the assumption based on the descriptors briefly, preferably in one or two sentences, in English.
3. You may need to focus on the implicit connections between their descriptors' conceptual space.
4. Based on the descriptors' conceptual space, you may speculate or assume what exactly the user preference or the item genres is.
5. Your explanation is based on the descriptors and your speculation.
Request:
1. Don't mention the name of the item (book title).
2. Don't try to understand every descriptor, just focus on the most relevant ones or the collective meaning of the descriptors.
3. Don't list the descriptors in the explanation, and don't use the word "descriptors".
4. Don't make general statements.
5. Just output the explanation in one or two sentences. No additional information is required, such as reasoning processes, evidence or notes.
### Input
Generate the interaction explanation for the following user and item based on their descriptors.
User descriptors: *<user descriptors>*.
Item descriptors: *<item descriptors>*.

---

Based on RLMRec profiles, the prompt template is as follows:

---

**Prompt for *Interaction Interpretation* based on RLMRec Profiles**

### Instruction
Act as an explanation generator for recommendation systems. Given a pair of user and item profiles, you need to generate a short explanation of why the user would like the item.
Requirements:
1. Assume that the user will interact with (or like) the item.
2. Provide your evidence to support the assumption based on the profiles briefly, preferably in one or two sentences.
3. Just explain the reason for the interaction, and don't make general statements.
4. The given user and item are from Amazon dataset, where the user and item are reader and book respectively.
5. DON'T mention the item's name (i.e. the title of the book) in the explanation. Use "book" instead (without quotes).
 ### Input
Generate the interaction explanation for the following user and item based on their profiles.
User profile: *<user profile>*.
Item profile: *<item profile>*.

---

After generating the explanation, we ask volunteers as well as LLMs to evaluate the quality of the explanation. We design a ranking setting where they are asked to rank the explanations for the 4 user-item pairs. The ranking is based on the quality of the explanation, where 1 is the best and 4 is the worst. For LLM ranking, we prompt the LLM to act as a recommender and provide a ranking for the explanations. The prompt template is as follows (same for explanations based on FACE descriptors and RLMRec profiles):

### Instruction
Data Description:
For each user corresponding to 4 items, only 1 item is a positive sample that interacts with the user, while the others are negative samples.
The explanation assumes that the user and the item will interact, and is generated by an LLM based on the descriptors or profiles of the user and item.
Focus on the credibility of the explanation to make judgments.
Task Requirement:
Rank the credibility of the interaction explanations for the 4 items of a user, with ranks from 1 to 4. The higher the rank, the more credible the explanation, and the more likely the user-item pair is a positive sample.
Output Format:
Output the four ranking numbers separated by spaces, without any other content, e.g., 1 2 3 4
### Input
For user u:
Item 1 Explanation: *<item1 explanation>*
Item 2 Explanation: *<item2 explanation>*
Item 3 Explanation: *<item3 explanation>*
Item 4 Explanation: *<item4 explanation>*
Please rank the explanations from 1 to 4 based on their credibility.

# F   Cold-start Experiments

Our framework provides an effective cold-start solution for collaborative filtering models. In FACE, when using an AutoEncoder for disentangled mapping, our model simultaneously trains a decoder that generates original collaborative representations from descriptors. We can leverage textual information from the dataset to generate descriptors for cold-start items. These descriptors can then be fed into the well-trained decoder to generate collaborative representations for cold-start items. The same principle applies to cold-start users.

We conducted an item zero-shot experiment to illustrate the model's cold-start performance: For a given dataset, 1/5 of the items were designated as cold-start items and excluded during training. During testing, we utilized the textual summary information to allow the LLM to generate descriptors, mimicking the association between textual information and descriptors of other items. The embeddings of these generated descriptors were then fed into the decoder to produce collaborative representations. Item recommendations were then made by calculating the similarity with user representations. We compared our approach against AlphaRec, a model with strong cold-start performance, using the same base model and textual information. The results are presented below:

Table 9: Cold-start performance comparison.

| Dataset | Metric | AlphaRec | FACE |
|---------|--------|----------|------|
| Amazon-book | R@5 | 0.0505 | 0.0630 |
| | R@20 | 0.1370 | 0.1395 |
| | N@5 | 0.0456 | 0.0611 |
| | N@20 | 0.0750 | 0.0912 |
| Yelp | R@5 | 0.0347 | 0.0342 |
| | R@20 | 0.1116 | 0.1010 |
| | N@5 | 0.0368 | 0.0374 |
| | N@20 | 0.0621 | 0.0658 |
| Steam | R@5 | 0.0447 | 0.0496 |
| | R@20 | 0.1287 | 0.1358 |
| | N@5 | 0.0430 | 0.0562 |
| | N@20 | 0.0729 | 0.0838 |

## G Complexity Analysis

Let $B = |\Omega|$ denote the batch size, $n$ the number of disentangled descriptors, $c = |\mathcal{D}|$ the size of the codebook for quantization, and $\bar{d}$ the embedding dimension. The disentanglement module, involving a disentangled mapping and a Transformer encoder, has a time complexity of $O(B(n^2\bar{d} + \bar{d}^2 n))$; The quantization module requires a nearest-neighbor search over the codebook, with a time complexity of $O(Bcn\bar{d})$; The Alignment stage, using a Transformer-based language model and contrastive loss, has a time complexity of $O(B(n^2\bar{d} + \bar{d}^2 n) + B^2\bar{d})$. Considering that the batch size $B$ is much smaller than the total number of users and items $N = |\mathcal{U}| + |\mathcal{I}|$, the additional computational cost of the FACE framework is linear with respect to $N$.

## H Limitations

Due to the computational resource constraints during experimentation, the current system adopts relatively small-scale language models (e.g., Llama2-7B, MiniLM-L6, Qwen2-7B); the framework's performance when integrated with larger-scale language models remains unverified. While the effectiveness of the smaller embedding model has been empirically validated in various studies, we hypothesize that adopting models with expanded parameter sizes could further enhance text embedding quality and vocabulary representation capabilities. In future work, we intend to extend our methodology to larger open-source LLMs (e.g., LLaMA-3 series or 70B-scale architectures) to fully evaluate the potential of sentence embedding and semantic alignment. Besides, our framework is currently limited to the collaborative filtering domain, and its applicability to other recommendation scenarios (e.g., sequence recommendation) remains unexplored. We plan to investigate the extension of our framework to these domains in future research.

## I Broader Impacts

Our proposed framework, FACE, aims to enhance the performance and interpretability of collaborative filtering recommendation systems via mapping CF embeddings into LLM tokens. It is a model-agnostic framework that can be applied to various CF models and LLMs, making it versa tile and adaptable to different recommendation scenarios. The interpretability aspect of FACE can help users understand the reasoning behind recommendations, potentially increasing user trust and satisfaction. Beyond improving recommendation accuracy and interpretability, convert CF embeddings into LLM tokens can also facilitate more downstream tasks. For instance, the generated descriptors can be fed into LLMs to carry out recommendation tasks, especially for cold-start scenarios, and descriptors can be modified specifically to achieve controllable recommendations.

However, despite the advancements offered by FACE, it is essential to acknowledge the potential drawbacks. The reliance on LLMs may introduce biases present in the original training data, which could lead to skewed recommendations as well as the bias in the generated explanations. Additionally, the computational resources required for LLMs may limit the accessibility of this approach for smaller organizations or researchers with limited resources.

