# OpenReview forum: "FACE: A General Framework for Mapping Collaborative Filtering Embeddings into LLM Tokens"
_NeurIPS.cc/2025/Conference — NeurIPS 2025 poster_

### Official Review · Reviewer_YJtU · 2025-06-29

**Clarity:** 3
**Significance:** 1
**Originality:** 2
**Rating:** 4
**Confidence:** 2

**Summary:**

This paper aims to bridge the semantic gap between the collaborative filtering models and the large language models. The authors proposed to map the CF ID embeddings into the semantic tokens of LLMs, with the following novelties: a disentangled projection module, a vector quantized variational autoencoder, and a contrastive alignment loss.

**Questions:**

1. The authors aim to achieve disentanglement when mapping the CF representation to the vocabulary space by employing "orthogonally initialized" projection heads. However, I'm not sure whether this (orthogonally initialized heads) is enough. These initially orthogonal heads may not become as orthogonal after the model optimization. I agree that disentangling these heads is beneficial, and I wonder whether it's better to achieve this goal by a more direct approach: employing a disentanglement loss (such as the de-correlation loss [1,2]) upon the outputs of these heads.

2. In the "Residual quantization" paragraph, the authors propose to "shares the same codebook across all quantization levels, with this codebook being composed of pre-trained LLM token embeddings". However, I'm not sure whether this is the best choice. The codebooks and "their embeddings" in RQ-VAE should contain a hierarchical structure of the data, while the LLM token embeddings don't. More experiments or analysis on justifying the claim would be great.

[1] Barlow twins: Self-supervised learning via redundancy reduction. PMLR, 2021.

[2] Enhancing CTR Prediction with De-correlated Expert Networks. 2025.

**Ethical Concerns:**

["NO or VERY MINOR ethics concerns only"]

**Final Justification:**

The authors provided detailed responses to address my questions regarding performance evaluation, analysis, and justification of the disentangled heads and codebooks. After resolving these concerns, I'm convinced of the novelty and the technical contribution of this paper.

**Limitations:**

1. Performance Evaluation. There are many existing works to inject LLM into recommendations(here I list several works for sequential recommendation:  [1,2,3], there should be many more for the general collaborative filtering/recommendation domain). In the performance evaluation, the authors should prove the effectiveness of the proposed approach by comparing it with these "LLM4Rec" methods, instead of just showing that FACE can enhance the performance of SOTA collaborative filtering models, because it's already well known that involving LLMs can boost the performance of CF. The authors' finding that "the gains achieved by the PLMRec model integrated with the FACE framework are minimal" precisely reflects my above-mentioned concern: existing work can already combine the LLM well, if so, then what's the advantage of FACE itself?

2. In-depth Analysis. Besides the performance evaluation, the author should provide more in-depth analysis on the advantages of the proposed approach over existing works. For example, does FACE align the two spaces better than existing works[1]?

3. Justification of the modules. The authors proposed several novel modules in Sec. 3.2 and 3.3. It would be great to justify them by: 1. What's the challenge the authors would like to tackle? 2. To what extent do existing approaches or naive methods resolve it? 3. Can the proposed module do better? How to measure or quantify it?

[1] CTRL: ConnectCollab- orative and Language Model for CTR Prediction. 2023.

[2] Enhancing Taobao Display Advertising with Multimodal Representations: Challenges. 2024.

[3] DisCo: Towards Harmonious Disentanglement and Collaboration between Tabular and Semantic Space for Recommendation. KDD 2024.

**Quality:**

2

**Strengths And Weaknesses:**

- Motivation. The authors aims to bridge the gap between the collaborative filtering and large language models. This is an interesting and critical research problem.

- Presentation. The whole paper is written well and is easy to follow.

- Reproduction. The authors provide the released code on the proposed FACE model for reproduction.

---

> ### Author Rebuttal · Authors · 2025-07-31
>
> We appreciate your comments. However, we need to emphasize that the main motivation and contribution of our paper.
>
> We are the first to **map hidden embeddings given by collaborative filtering methods into pre-trained LLM tokens**, in order to improve the **interpretability** of arbitrary traditional CF models, and to this end, we propose a general and interpretable semantic mapping framework. This framework enables large language models to understand recommendation embeddings through semantic mapping, thereby improving interpretability as well as recommendation performance by leveraging LLM knowledge. For this, we have designed extensive experiments to demonstrate the quality of the mapped tokens and the interpretability beyond recommendation performance.
>
> > Response to Q1:
>
> Thank you for your suggestions. In fact, orthogonally initialization is enough for the projection heads. Our model inherently tends to capture comprehensive preferences, automatically forcing disentanglement heads to capture different aspects, as the multiple mapped descriptors are required to reconstruct the original representation and align with semantic information. For validation, we design an experiment. We have implemented the de-correlation loss from the your listed method [2] to constrain the outputs of the multi-projectors. The result could be because it is difficult to balance this loss with the other losses, or this additional constraint hinders the semantic alignment.
>
> |                | Amazon     |            |            |            | Yelp       |            |            |            |
> | -------------- | ---------- | ---------- | ---------- | ---------- | ---------- | ---------- | ---------- | ---------- |
> |                | R@5        | R@20       | N@5        | N@20       | R@5        | R@20       | N@5        | N@20       |
> | de-correlation | 0.0698     | 0.1609     | 0.0696     | 0.0993     | 0.0435     | 0.1184     | 0.0507     | 0.0749     |
> | FACE           | **0.0705** | **0.1622** | **0.0705** | **0.1009** | **0.0446** | **0.1203** | **0.0519** | **0.0766** |
>
> > Response to Q2:
>
> In the original paper of RQ-VAE (Lee et al. 2022), the author proposed and adopted shared codebook across all quantization levels. This technique maximize the utilization of codebook, making "all code embeddings available for every quantization depth". And we follow this setting in our work. For our work, although LLM token embeddings do not have an explicit hierarchical structure, the vector arithmetic of their embeddings is meaningful (e.g., king - man + woman ≈ queen). When obtaining the final quantized vector $z_q = \sum_{i=1}^H c^{(h)}$, this property can be leveraged. Additionally, due to the nature of residual quantization, the first-level quantization result for each disentangled aspect captures the most significant information, which is then used for the interpretability experiments. For example, SPAE (Yu et al., 2023)  in the image-text multi-modal domain, also employs the same LLM vocabulary for its multi-level quantization.
>
> > Response to L1:
>
> First, to the best of our knowledge, our work is the first to map collaborative representations to **pre-trained**, **fixed** semantic tokens in the LLM for interpretability and is **not merely for improving recommendation performance using textual information**. Such a general framework also holds significant potential for many downstream tasks, such as conducting interpretable analysis, generating embeddings for cold-start items using the decoder, or leveraging the controllable recommendation capabilities of LLMs after mapping users and items to tokens.
>
> Second, our framework can be applied to RLMRec, a collaborative filtering model that already incorporates textual information, to achieve further improvements in performance, even if the improvement is less compared to its application on traditional recommendation models that do not use textual information. This is the reason why we wrote, "the gains achieved by RLMRec model integrated with FACE are minimal," and we apologize if our imprecise wording caused any misunderstanding.
>
> To fairly compare the ability of different frameworks to align with large language models, it is necessary to conduct experiments under the same base model (LightGCN) and with the same textual information. We have included the method you listed, CTRL (Wang et al., 2023), which aims at CTR prediction in collaborative filtering. Additionally, we have included another enhancement framework that utilizes large models and text, KAR (Xi et al., 2023), and we simultaneously investigate the effect of further applying FACE to these LLM-enhanced recommendation models.
>
> |                   | Amazon     |            |            |            | Yelp       |            |            |            | Steam      |            |            |            |
> | ----------------- | ---------- | ---------- | ---------- | ---------- | ---------- | ---------- | ---------- | ---------- | ---------- | ---------- | ---------- | ---------- |
> |                   | R@5        | R@20       | N@5        | N@20       | R@5        | R@20       | N@5        | N@20       | R@5        | R@20       | N@5        | N@20       |
> | LightGCN          | 0.0659     | 0.1563     | 0.0657     | 0.0961     | 0.0421     | 0.1141     | 0.0488     | 0.0726     | 0.0530     | 0.1361     | 0.0584     | 0.0862     |
> | **LightGCN+FACE** | 0.0705     | **0.1622** | 0.0705     | 0.1009     | 0.0446     | 0.1203     | **0.0519** | 0.0766     | 0.0559     | 0.1439     | 0.0611     | 0.0912     |
> | RLMRec            | 0.0669     | 0.1572     | 0.0663     | 0.0981     | 0.0426     | 0.1165     | 0.0495     | 0.0737     | 0.0545     | 0.1408     | 0.0599     | 0.0887     |
> | **RLMRec+FACE**   | 0.0679     | 0.1581     | 0.0672     | 0.0985     | 0.0435     | 0.1196     | 0.0503     | 0.0755     | 0.0556     | 0.1432     | 0.0604     | 0.0901     |
> | CTRL              | 0.0685     | 0.1566     | 0.0666     | 0.0963     | 0.0413     | 0.1161     | 0.0471     | 0.0719     | 0.0531     | 0.1375     | 0.0586     | 0.0865     |
> | **CTRL+FACE**     | 0.0699     | 0.1612     | 0.0706     | **0.1013** | 0.0432     | 0.1189     | 0.0510     | 0.0755     | **0.0569** | **0.1461** | **0.0620** | **0.0922** |
> | KAR               | 0.0671     | 0.1547     | 0.0660     | 0.0952     | 0.0408     | 0.1132     | 0.0476     | 0.0712     | 0.0502     | 0.1322     | 0.0553     | 0.0826     |
> | **KAR+FACE**      | **0.0712** | 0.1559     | **0.0714** | 0.1002     | **0.0446** | **0.1249** | 0.0518     | **0.0781** | 0.0516     | 0.1349     | 0.0571     | 0.0852     |
>
> The experimental results demonstrate that: 1. FACE achieves superior performance compared to other LLM-enhanced recommendation methods, meaning that LightGCN, which does not inherently use semantic information, outperforms existing LLM4Rec methods when integrated with FACE; 2. FACE can be effectively applied to various existing LLM-enhanced recommendation approaches to further improve their performance. These findings are consistent with those presented in the original paper, highlighting the advantages of the FACE framework in utilizing semantic mapping to enable LLMs to comprehend collaborative embeddings.
>
> > Response to L2:
>
> Our work focuses on mapping hidden embeddings given by collaborative filtering methods into interpretable LLM tokens, which to our knowledge has not been explored. Existing LLM4Rec alignment works directly align collaborative space with textual space by contrastive learning merely for textual injection and performance improvement (RLMRec, CTRL, etc.), while our work enables LLM to interpret collaborative embeddings and improve interpretability. Therefore, besides the performance evaluation, we have designed various interpretability experiments (Section 4.4) to demonstrate the quality and the application of semantic mapping.  For interpretability, we design item-retrieval, item-generation tasks and real-user study on LLM interpretation on interactions. These studies quantify the quality of semantic mapping, and suggest that our method constructs effective semantic mapping and improve the interpretability of latent CF embedding.
>
> > Response to L3:
>
> + **What's the challenge the authors would like to tackle?** The challenge we aim to tackle is the semantic interpretability of user and item embeddings in traditional collaborative filtering models. To address this, we propose a general semantic mapping framework that is applicable to any CF model and any LLM.
>
> + **To what extent do existing approaches or naive methods resolve it?** Traditional models do not address this issue at all. Most existing interpretable methods use language models to explain *interaction outcomes*, but they do not focus on providing interpretability at the embedding level. Knowledge-enhanced methods, on the other hand, typically leverage semantics solely to improve the recommendation performance of collaborative filtering, not for interpretability.
>
> + **Can the proposed module do better? How to measure or quantify it?** Given that we are the first to tackle the semantic interpretability of user and item embeddings in traditional collaborative filtering models, besides recommendation performance evaluation, we designed various experiments to specifically quantify interpretability. These include item-retrieval and item-generation tasks, as well as a real-user study on LLM interpretation of interactions. These studies quantify the quality of the semantic mapping and suggest that our method improves the interpretability of latent CF embeddings by constructing this mapping.

---

> ### Author Response · Authors · 2025-08-01
>
> Dear Reviewer,
>
> We greatly appreciate the time and effort you have invested in evaluating our work. We have uploaded a detailed rebuttal addressing all the concerns and questions raised in your reviews.
>
> We would like to clarify a minor phrasing issue in our original rebuttal. In the following passage: "**Can the proposed module do better? How to measure or quantify it?** Given that we are the first to tackle the semantic interpretability of user and item embeddings in traditional collaborative filtering models, besides recommendation performance evaluation, we designed various experiments to specifically quantify interpretability." We meant to emphasize that we are the first to tackle CF embedding interpretability through LLM vocabulary mapping. Our key contribution lies in proposing the novel approach of using pre-trained LLM token vocabulary to semantically interpret collaborative filtering embeddings, which represents a fundamentally different methodology from existing interpretability approaches.
>
> Thank you for your understanding and we look forward to your feedback.

---

> ### Author Response · Authors · 2025-08-05
>
> Dear Reviewer,
>
> Thank you for taking the time to review our paper. We have provided detailed responses to address your concerns and questions. Please let us know if there are any remaining points that need clarification - we would be happy to discuss them further.
>
> Best regards,

---

> ### Author Response · Authors · 2025-08-07
>
> Dear Reviewer YJtU,
>
> Thank you for your review. We have provided a detailed rebuttal addressing all your concerns, including:
>
> 1. Clarifying that FACE is the first framework to map CF embeddings into pre-trained LLM tokens for interpretability (not just another "inject LLM into recommendations" approach)
> 2. Providing additional experiments comparing with CTRL and KAR methods
> 3. Addressing your questions about disentanglement and quantization design choices
>
> We would appreciate your thoughts on our responses, especially regarding the fundamental contribution of our semantic mapping framework. With the discussion deadline approaching, we hope to engage in productive dialogue to clarify any misunderstandings.
>
> Thank you for your consideration.
>
> Best regards,

---

> > ### Comment · Reviewer_YJtU · 2025-08-07
> >
> > Thanks for the authors' detailed response, which addresses most of my concerns. Please add these additional results in the revised version. I will raise my rating score to a positive one.

---

> > > ### Author Response · Authors · 2025-08-07
> > >
> > > Dear Reviewer YJtU,
> > >
> > > Thank you very much for your positive feedback and for raising your rating. We truly appreciate your engagement in the discussion.
> > >
> > > We will definitely add all the additional experimental results in the revised version as promised.
> > >
> > > Thank you again for your valuable comments.
> > >
> > > Best regards,

---

### Official Review · Reviewer_yz2S · 2025-06-30

**Clarity:** 3
**Significance:** 4
**Originality:** 4
**Rating:** 5
**Confidence:** 5

**Summary:**

This paper proposes FACE, a general framework for mapping collaborative filtering (CF) embeddings into interpretable LLM tokens to bridge the gap between non-semantic CF representations and LLM understanding. The motivation stems from the challenge that LLMs cannot directly interpret latent CF embeddings, limiting their effectiveness in recommendation systems. The methodology involves two main components: (i) a disentangled projection module that decomposes CF embeddings into concept-specific vectors, and (ii) a quantized autoencoder that converts continuous embeddings into discrete LLM tokens (descriptors) using residual quantization with a frozen LLM vocabulary as codebook. A contrastive alignment objective ensures semantic consistency between mapped tokens and textual signals. Experiments on three datasets demonstrate consistent performance improvements across various CF models, with interpretability studies confirming that the generated descriptors can be understood by LLMs.

**Questions:**

1. Could the authors provide more theoretical or empirical analysis on how the semantic alignment objective (contrastive learning) directly contributes to recommendation accuracy improvements, beyond just enabling interpretability? The connection between these dual objectives needs clearer justification.

2. How does FACE compare with other recent LLM-enhanced recommendation methods beyond RLMRec? Additionally, could the authors discuss the computational overhead and scalability of the three-stage training process compared to simpler alternatives?

3. Can the trained FACE framework be transferred across different domains or datasets without retraining? What are the limitations regarding domain adaptation, and how sensitive is the method to the quality and availability of textual features in new datasets?

**Ethical Concerns:**

["NO or VERY MINOR ethics concerns only"]

**Final Justification:**

The authors have further addressed my concerns mentioned in the weakness part. I believe this is a clear accept paper given its novelty.

**Limitations:**

yes

**Quality:**

3

**Strengths And Weaknesses:**

**Strengths:**
- The paper targets a useful and interesting research task in LLM-enhanced recommendation. How can we translate hidden embeddings given by collaborative filtering models into understandable LLM tokens? The paper decomposes this task into multiple steps: i) disentangling CF embeddings into multiple embeddings related to specific semantic aspects, and ii) quantizing the embeddings into LLM tokens. The framework design is elegant and intuitive, addressing a fundamental challenge in bridging CF and LLM modalities. This design not only greatly improves the explainability of recommendation models, but also provides a better alignment with textual features to improve recommendation accuracy.

- The novelty of the proposed method is notable, particularly in the quantization approach that uses frozen LLM vocabulary as codebook combined with residual quantization for hierarchical semantic representation. The disentanglement method through orthogonally initialized multi-projectors followed by transformer encoding is also innovative. The integration of these components with contrastive alignment learning represents a significant technical contribution that enables end-to-end learning of semantic mappings.

- The proposed method provides consistent performance improvements for recommendation models of different categories, demonstrating its model-agnostic nature and general applicability across various CF architectures from matrix factorization to graph neural networks.

- The real user study effectively demonstrates the interpretability of the proposed method, with both item retrieval and generation tasks showing that LLMs can understand and utilize the generated descriptors, and human evaluation confirming the quality of interaction explanations.

- The paper is well-written with clear motivation, comprehensive methodology description, and thorough experimental evaluation. The technical details are presented clearly, and the experimental setup is comprehensive with proper ablation studies and hyperparameter analysis.

**Weaknesses:**
- The proposed method is mainly focused on improving the explainability of recommendation models, but the experiments are mainly designed to evaluate recommendation accuracy. There is a gap in the former part of the paper to encounter the connection between these two objectives.

- The proposed method only involves one LLM-enhanced recommendation model (i.e. RLMRec) in the experiments, limiting the comparison with other LLM-based recommendation approaches and making it difficult to assess the relative advantages of FACE over alternative LLM integration strategies.

- The proposed method seems reliant on the training dataset (textual features and the results of CF training on this dataset). It is not clear how the trained mapping functions can be applied to other datasets or domains, raising questions about the transferability and generalizability of the learned mappings.

---

> ### Author Rebuttal · Authors · 2025-07-31
>
> We sincerely thank you for your recognition of our work, and we are encouraged by your recognition that “The novelty of the proposed method is notable”. Here are our replies to your concerns.
>
> > Response to W1: About dual objectives.
>
> We are the first to propose a novel semantic mapping method that creatively maps collaborative filtering representations into LLM tokens. This direct mapping improves explainability as well as accuracy. **The dual objectives of improving explainability and accuracy are not only the results of this semantic mapping but also serve to validate its effectiveness in turn.** The improvement in recommendation performance provides initial evidence that the mapped tokens can be understood by the LLM, which leverages its textual knowledge to enhance model performance. Furthermore, we conducted extensive experiments, including item generation/retrieval and real-user ranking studies, to further demonstrate that the descriptors and the relationships between user/item descriptors can be comprehended by the LLM, which enhances the interpretability of the embedding-based collaborative filtering method.
>
> > Response to Q1: How the semantic alignment objective (contrastive learning) directly contributes to recommendation accuracy improvements?
>
> Regarding the direct contribution of semantic alignment to accuracy, as demonstrated in RLMRec (Xi et al., 2023), maximizing the "hidden prior belief" beneficial for recommendation is equivalent to maximizing **the mutual information between the CF embedding and the semantic representation**. Our method can be viewed as a specific approach to maximizing this mutual information. Given the design of our semantic alignment, the alignment loss signal can be learned by the initial CF embeddings, thereby improving recommendation performance.
>
> Empirically, our **ablation studies** in Section 4.6 (the "w/o align" variant) confirms that the accuracy improvement stems from semantic alignment. Additionally, the **hyperparameter sensitives** in Section 4.5 on the weight of the alignment loss shows that this parameter controls the intensity of semantic alignment (i.e., the amount of textual information injected). The recommendation metrics exhibit a trend of first increasing, linking the alignment objective to performance gains, and then decreasing as this coefficient rises, demonstrating too much information injection affects recommendation.
>
> > Response to W2 & Q2: How does FACE compare with other recent LLM-enhanced recommendation methods beyond RLMRec?
>
> We have conducted new experiments comparing FACE with other recent LLM-enhanced recommendation methods:  CTRL(Wang et al. 2023) leverages contrastive learning for fine-grained knowledge alignment and integration with collaborative and language model; KAR (Xi et al., 2023) combine LM representations with traditional recommenders using a hybrid-expert adaptor. We compared all methods using the same textual embeddings and LightGCN as the backbone. The new experimental results demonstrate that: 1. FACE achieves superior performance compared to other LLM-enhanced recommendation methods; 2. FACE can be effectively applied to various existing LLM-enhanced recommendation approaches to further improve their performance.  These findings are consistent with those presented in our original paper, that proves the advantage of FACE.
>
> |                   | Amazon     |            |            |            | Yelp       |            |            |            | Steam      |            |            |            |
> | ----------------- | ---------- | ---------- | ---------- | ---------- | ---------- | ---------- | ---------- | ---------- | ---------- | ---------- | ---------- | ---------- |
> |                   | R@5        | R@20       | N@5        | N@20       | R@5        | R@20       | N@5        | N@20       | R@5        | R@20       | N@5        | N@20       |
> | LightGCN          | 0.0659     | 0.1563     | 0.0657     | 0.0961     | 0.0421     | 0.1141     | 0.0488     | 0.0726     | 0.0530     | 0.1361     | 0.0584     | 0.0862     |
> | **LightGCN+FACE** | 0.0705     | **0.1622** | 0.0705     | 0.1009     | 0.0446     | 0.1203     | **0.0519** | 0.0766     | 0.0559     | 0.1439     | 0.0611     | 0.0912     |
> | RLMRec            | 0.0669     | 0.1572     | 0.0663     | 0.0981     | 0.0426     | 0.1165     | 0.0495     | 0.0737     | 0.0545     | 0.1408     | 0.0599     | 0.0887     |
> | **RLMRec+FACE**   | 0.0679     | 0.1581     | 0.0672     | 0.0985     | 0.0435     | 0.1196     | 0.0503     | 0.0755     | 0.0556     | 0.1432     | 0.0604     | 0.0901     |
> | CTRL              | 0.0685     | 0.1566     | 0.0666     | 0.0963     | 0.0413     | 0.1161     | 0.0471     | 0.0719     | 0.0531     | 0.1375     | 0.0586     | 0.0865     |
> | **CTRL+FACE**     | 0.0699     | 0.1612     | 0.0706     | **0.1013** | 0.0432     | 0.1189     | 0.0510     | 0.0755     | **0.0569** | **0.1461** | **0.0620** | **0.0922** |
> | KAR               | 0.0671     | 0.1547     | 0.0660     | 0.0952     | 0.0408     | 0.1132     | 0.0476     | 0.0712     | 0.0502     | 0.1322     | 0.0553     | 0.0826     |
> | **KAR+FACE**      | **0.0712** | 0.1559     | **0.0714** | 0.1002     | **0.0446** | **0.1249** | 0.0518     | **0.0781** | 0.0516     | 0.1349     | 0.0571     | 0.0852     |
>
> > Response to Q2: Discuss the computational overhead and scalability of the three-stage training process compared to simpler alternatives.
>
> At first, we conduct a time complexity analysis of the proposed framework. Let $B$ denote the batch size, $D$ the number of disentangled descriptors, $C$ the size of the codebook for quantization, and $d$ the embedding dimension. The disentanglement module, involving a disentangled mapping and a Transformer encoder, has a time complexity of $O(B(D^2d + d^2D))$; The quantization module requires a nearest-neighbor search over the codebook, with a time complexity of $O(BCDd)$; The Alignment stage, using a Transformer-based language model and contrastive loss, has a time complexity of $O(B(D^2d + d^2D) + B^2d)$.
>
> Considering that the batch size $B$ is much smaller than the total number of users and items $N=|\mathcal U|+|\mathcal I|$, **the additional computational cost of the FACE framework is linear with respect to $N$**. Additionally, the number of descriptors $D$ is typically set to a small constant, the actual increase in computational load is modest.
>
> **The three-stage training strategy was designed not only for better convergence but also to reduce computational time.** In the first stage, we train the base model. The simple yet effective LightGCN model has a training complexity of $O(L⋅|\mathcal E|⋅d)$, where $|\mathcal E|$ is the number of interactions and $L$ the number of convolution layers. In the second stage, we only combine the AutoEncoder, without alignment. The full framework is trained in the final stage, which converges in just a few epochs due to effective pre-training—significantly faster than training the whole model from scratch. The time complexity of all the three stage is linear with respect to $|\mathcal U|, |\mathcal I|$ or $|\mathcal E|$ and is efficient in practical scenario where they grow rapidly.
>
> For comparison, RLMRec based on LightGCN, introduces an additional complexity of $O(Bd^2 + B^2d)$, quadratic with respect to the batch size $B$, but the overall training overhead remains linear with respect to the total number of users/items $N$, similar to our FACE framework.
>
> > Response to W3 & Q3: Can the trained FACE framework be transferred across different domains or datasets without retraining?
>
> Thank you for this insightful question regarding the framework's transferability. We propose a model-agnostic framework that can be integrated with various collaborative filtering methods. The transferability of FACE is contingent upon the capabilities of the underlying base model. If the base model supports cross-domain recommendation, FACE may inherit this transferability. We will continue to explore the performance of the FACE framework in cross-domain settings in our future work.

---

> > ### Comment · Reviewer_yz2S · 2025-08-08
> >
> > Thank you to the authors for your comprehensive response. I see that most reviewers have given high ratings. I believe this is a clear accept paper given its novelty.

---

> > > ### Author Response · Authors · 2025-08-09
> > >
> > > Dear Reviewer yz2S,
> > >
> > > Thank you for your thoughtful review and positive feedback on our paper. We truly appreciate your time and valuable insights.
> > >
> > > Best regards,

---

### Official Review · Reviewer_SnQC · 2025-07-03

**Clarity:** 3
**Significance:** 3
**Originality:** 3
**Rating:** 5
**Confidence:** 5

**Summary:**

This paper presents **FACE**, a novel framework designed to bridge collaborative filtering (CF) models with large language models (LLMs) by mapping CF embeddings into LLM tokens. It introduces a disentangled projection module and a quantized autoencoder to convert CF embeddings into discrete LLM tokens. Additionally, a contrastive alignment objective is proposed to ensure semantic consistency throughout the process.

**Questions:**

See Weaknesses.

**Ethical Concerns:**

["NO or VERY MINOR ethics concerns only"]

**Final Justification:**

Based on the Strengths and the rebuttals, I suggest Accept for this paper.

**Limitations:**

I believe this work does not pose potential negative societal impacts.

**Quality:**

4

**Strengths And Weaknesses:**

**Strengths**

1. The methodology is well-structured and technically sound. The combination of disentangled projection, residual quantization, and contrastive learning reflects a thoughtful approach to addressing the complexities of mapping between different embedding spaces.

2. The authors provide anonymized codes, which enhances the reproducibility of the work.

3. Comprehensive experiments are conducted to effectively demonstrate the proposed method's efficacy.

**Weaknesses**

1. In Section 3.3.1, the description of the Descriptor Embedding is somewhat difficult to follow. It would be beneficial to provide a clearer explanation of "descriptor embeddings," including the distinction between descriptor and summary embeddings. Additionally, clarification on the pseudo-inverse matrix and how it is obtained would be helpful.

2. A more detailed time complexity analysis of the proposed framework would be advantageous.

---

> ### Author Rebuttal · Authors · 2025-07-31
>
> We sincerely thank you for your recognition of our work, noting that “The methodology is well-structured and technically sound.” Here are our replies to your concerns.
>
> > W1.1: It would be beneficial to provide a clearer explanation of "descriptor embeddings," including the distinction between descriptor and summary embeddings.
>
> Thank you for your suggestions. **Summary embeddings** are derived from textual information in the dataset, and then processed by LLM enhancement and embedding. In contrast, **descriptor embeddings** originate from the collaborative embeddings. Specifically, a collaborative embedding is mapped to a set of discrete "descriptors" (LLM tokens), denoted as $z_d$. These tokens are then inserted into a prompt template to form a complete sentence, which is subsequently fed into the LLM to generate a final, semantically rich sentence embedding. We will provide a clearer explanation in the revised version.
>
> > W1.2: Clarification on the pseudo-inverse matrix and how it is obtained would be helpful.
>
> Regarding the **pseudo-inverse matrix**: during quantization, we obtain the token embedding $c$ (i.e., $z_d$) after dimensionality reduction. It is required to recover the original token embedding $c_0$ while maintaining the gradient for the alignment stage. Considering the projection matrix $W_c \in \mathbb{R}^{d \times d_{LLM}}$, since $d<d_{LLM}$,, a standard inverse matrix does not exist. Therefore, we employ the pseudo-inverse to solve this problem. From the relation $c=W_cc_0$, **we can derive the solution for $c_0$ as**:
>
> $$c_0 =(W_c^TW_c)^{−1}W_c^T c$$ ,
>
> where we denote $(W^T_cW_c)^{-1}W_c^T$ as the pseudo-inverse matrix $W_c^{-1}$. While computing, this requires $W_c$ to have full row rank (to ensure $W_c^T W_c$ is invertible), this condition can be met in practice by adding a small perturbation to the weights for numerical stability. We really appreciate your suggestions, and will provide a clearer explanation in the revised version.
>
> > W2: A more detailed time complexity analysis of the proposed framework would be advantageous.
>
> Thank you for your valuable suggestions. Let $B$ denote the batch size, $D$ the number of disentangled descriptors, $C$ the size of the codebook for quantization, and $d$ the embedding dimension.
>
> + The **Disentanglement module**, involving a disentangled mapping and a Transformer encoder, has a time complexity of $O(B(D^2d + d^2D))$.
> + The **Quantization module** requires a nearest-neighbor search over the codebook, with a time complexity of $O(BCDd)$.
> + The **Alignment stage**, using a Transformer-based language model and contrastive loss, has a time complexity of $O(B(D^2d + d^2D) + B^2d)$.
>
> Considering that the batch size $B$ is much smaller than the total number of users and items $N$, the additional computational cost of the FACE framework is linear with respect to $N$. This is important in practical scenario where $N$ grows rapidly, which shows the efficiency of our proposed framework.

---

> > ### Comment · Reviewer_SnQC · 2025-08-03
> >
> > Thanks for the rebuttal. My concerns are addressed. I think this is a good paper.

---

> > > ### Author Response · Authors · 2025-08-03
> > >
> > > Dear Reviewer,
> > > Thank you very much for your valuable feedback.
> > > Best regards

---

### Official Review · Reviewer_kYKc · 2025-07-04

**Clarity:** 3
**Significance:** 3
**Originality:** 4
**Rating:** 5
**Confidence:** 5

**Summary:**

The paper proposes FACE, a model-agnostic recommendation framework that combines a RQ-VAE and contrastive learning to map collaborative filtering embeddings into a set of pretrained LLM tokens. It aims to leverage the power of LLMs to improve both performance and interpretability.

**Questions:**

+ Just an extended question: can the proposed autoencoder-based bidirectional mapping strategy be extended to facilitate alignment across other modalities?

**Ethical Concerns:**

["NO or VERY MINOR ethics concerns only"]

**Final Justification:**

Since the authors have addressed my concerns, I have decided to maintain my positive score and recommend accepting this paper.

**Limitations:**

yes

**Paper Formatting Concerns:**

There are no apparent formatting issues.

**Quality:**

4

**Strengths And Weaknesses:**

The core idea of enabling direct interpretation of collaborative filtering representations by large language models addresses a fundamental challenge in the field. This approach establishes a bridge between traditional CF methods and modern LLMs, which could potentially unlock new possibilities for advanced recommendation tasks such as controllable generation, cross-domain transfer, and multi-modal recommendation scenarios.

The paper has the following strengths:

S1: The proposed strategy of combining a quantized autoencoder for CF-LLM mapping while maintaining information integrity and contrastive learning for semantic alignment demonstrates solid technical grounding. The design choices appear well-motivated and the three-stage training strategy shows practical consideration for optimization stability.

S2: The framework's model-agnostic nature represents a significant practical advantage. Rather than requiring specific architectural modifications, FACE can be applied to various existing CF models (GMF, LightGCN, SimGCL, etc.), which enhances its potential for widespread adoption and real-world deployment.

S3: The direct mapping of embeddings into existing semantic tokens as descriptors offers an elegant solution to the interpretability challenge. This approach not only provides human-readable explanations but also enables LLMs to perform complex reasoning tasks based on these semantic representations, opening doors for sophisticated recommendation applications beyond simple ranking.

S4: The experimental validation is comprehensive, demonstrating consistent improvements across multiple datasets and base models. The interpretability studies, particularly the real user evaluation comparing FACE descriptors with RLMRec profiles, provide convincing evidence that the mapped tokens capture meaningful semantic information while being more concise (16 tokens vs. paragraph-length profiles).

The paper may have the following weaknesses:

W1: While the overall methodology is well-structured, the distinction between the latent variables $z_e$, $z_d$, and $z_q$ could be clarified further. For instance, by emphasizing the role of $z_d$ or providing a brief background on RQ-VAE to aid reader understanding.

w2: Some text in Figure 1 is a bit small, which could be adjusted to improve readability for readers.

Overall, I think this paper presents very interesting and beneficial work for recommendation systems. Based on the above analysis, I recommend acceptance of this paper.

---

> ### Author Rebuttal · Authors · 2025-07-31
>
> We sincerely thank you for your insightful feedback. We are particularly encouraged by your recognition of our work's significance: "The core idea of enabling direct interpretation of CF representations by LLM addresses a fundamental challenge in the field." Below, we address your suggestions and questions.
>
> > On Clarification and Readability (W1, W2)
>
> We appreciate your constructive suggestions regarding clarity. In the revised manuscript, we will incorporate the following changes:
>
> **Clarifying Variables in RQ**: We will expand Section 3.2.2 and the caption of Figure 1 to further clarify the roles of the latent variables.
>
> + $z_e$: The disentangled collaborative embedding derived from the CF model.
>
> + $z_d$: The quantized embedding corresponding to a descriptor token from the LLM's vocabulary. This variable plays a pivotal role in achieving semantic alignment and interpretability, as the alignment loss is backpropagated through $z_d$.
> + $z_q$: The approximate disentangled collaborative embedding, constructed from the sum of different levels of quantization codewords.
>
> **Improving Figure Readability**: We will increase the font size of the text in Figure 1 to ensure better readability for all readers.
>
> > Can the proposed strategy be extended to facilitate alignment across other modalities? (Q1)
>
> Thank you for this question. We believe the core principles of our framework are generalizable. Our method establishes a bidirectional mapping between continuous representations (from CF models) and discrete semantic tokens (from an LLM). The efficacy of this mapping is ensured by the autoencoder's reconstruction architecture and continuous-to-discrete quantization, while semantic alignment is achieved via contrastive learning. Crucially, we maintain effective gradient flow from the discrete token space back to the continuous embedding space by using a pseudo-inverse mapping. This set of techniques is not inherently limited to recommendation systems. It could potentially be adapted for other cross-modal alignment tasks, such as aligning continuous image embeddings with discrete textual tokens. This is an exciting direction for future work that we are keen to explore. Thank you again for your valuable time and thoughtful review.

---

> > ### Comment · Reviewer_kYKc · 2025-08-03
> >
> > Thank you for the authors' detailed response, which has adequately addressed my concerns. I will maintain my positive score.

---

> > > ### Author Response · Authors · 2025-08-03
> > >
> > > Dear Reviewer,
> > > Thank you very much for your valuable feedback.
> > > Best regards

---

### Official Review · Reviewer_dB6G · 2025-07-04

**Clarity:** 3
**Significance:** 2
**Originality:** 3
**Rating:** 3
**Confidence:** 4

**Summary:**

The paper proposes a tokenization framework FACE, that bridges the gap between embedding-based representation derived from collaborating filtering methods and the discrete representations used in LLM embeddings. The method maps the CF embeddings into discrete LLM tokens referred to as descriptors. To capture multiple semantically meaningful concepts, the method leverages disentangled projections of the CF embedding and then learns a RQ-VAE tokenizer with frozen LLM embeddings as the codebook. In addition, a contrastive alignment objective is introduced for learning semantically meaningful user/item representations, allowing LLMs to understand items/users without additional LLM fine-tuning. Experiments across three different recommendation datasets depict the efficacy of the method proposed.

**Questions:**

- In Section 3.3.1, it is stated that we have summaries for both users and items. Are we doing contrastive loss for both items and users separately? Please clarify in the text?
- For joint training, what is recommendation objective? Is it retrieval/ranking task? Do we optimize all the parameters of the LLM for this task?
- Why is Residual Quantization used if we only consider the first code as the descriptor of the encoded disentangled CF embedding?
- How does the method generalize in the cold-start setting for users and items?

**Ethical Concerns:**

["NO or VERY MINOR ethics concerns only"]

**Final Justification:**

Some of my concerns have been addressed. I greatly appreciate the new cold-start experiment that were included to show that this method can be used for cold-start generalization. I am still unsure about the scalability arguments (as discussed in my comment). I still think the pros outweigh the cons. Hence, the rating is increased to 4.

**Limitations:**

yes

**Quality:**

2

**Strengths And Weaknesses:**

**Strengths**
- The proposal for doing end-to-end tokenization with LLM finetuning for recommendation tasks seems novel.
- The paper is overall clear and easy to understand.

**Weaknesses**

*Lack of discussions around scalability*
- The key desirable property for models in recommendation system is the ability to scale and to be able to operate in cold-start settings. However, the paper doesn't discuss this. How will the chosen LLM codebook scale to millions/billions of items for a real-world recommendation system corpus? Can the alignment of CF embeddings and LLM  embeddings help with generalization in cold-start settings?

*Missing crucial baselines and citations*
- Generative Retrieval for RecSys (Rajput et al. 2023): This paper first introduced learning Semantic ID for recommendation items using RQ-VAE for the generative retrieval task. This paper should be among the main baselines for the proposed item tokenization approach.
- Traditional recommender systems (S3-Rec (Zhou et al. (2020)), SASRec (Kang et al. 2018)) are not considered as baselines.
- SPAE tokenization SPAE: Semantic Pyramid AutoEncoder for Multimodal Generation with Frozen LLMs. This work also uses frozen LLM embeddings to learn hierarchical descriptors when learning multi-modal tokens. Please consider adding it to related work.

*Few implementation details are unclear*
- In Section 3.3.1, it is stated that we have summaries for both users and items. Are we doing contrastive loss for both items and users separately? Please clarify in the text?
- For joint training, what is recommendation objective? Is it retrieval/ranking task? Do we optimize all the parameters of the LLM for this task?
- Why is Residual Quantization used if we only consider the first code as the descriptor of the encoded disentangled CF embedding?

*Unclear where the improvement is coming from*
- It is not clear from the paper what is causing the improvement shown in Table 1. Is it because of the proposed tokenization framework or  because the LLMs already have enough knowledge about these chosen datasets? Perhaps LLMs already contain the collaborative knowledge of these open-sourced recommendation benchmarks?

---

> ### Author Rebuttal · Authors · 2025-07-31
>
> Thank you for your valuable comments and for recognizing the novelty of our approach.
>
> However, first of all, we would like to clarify that the main contribution our proposed framework, FACE, is a **general** framework mapping collaborative embeddings from any CF model into pre-trained tokens from any LLM, which enhances **interpretability** as well as recommendation performance. This is the first general enhancement framework that enables LLMs to directly understand collaborative representations in the form of LLM tokens. Below are our responses to your concerns.
>
> > Response to W1 & Q4: scalability & cold-start
>
> The chosen LLM codebook in our framework has good **scalability**, and is enough for practical recommendation scenario. Let $|\mathcal{D}|$ denote the size of the LLM vocabulary, $n$ the number of disentangled descriptors, and $H$ the number of RQ layers. The number of possible token combinations reaches: $$C^n_{|\mathcal{D}|} \times (A^n_{|\mathcal{D}|})^{H-1} = \frac{(A^n_{|\mathcal{D}|})^{H}}{A^n_n}$$ Considering a semantic-filtered vocabulary size of approximately 2000 tokens, with $n=16$ disentangled descriptors and $H=3$ layers, this number is estimated to be over $10^{100}$. Even for $H=1$, it is approximately $10^{34}$. This capacity is sufficient to represent items in real-world recommendation scenarios. For complex or information-rich real-world item/user spaces, the number of tokens mapped per item can be increased for further scalability.
>
> Our framework provides **an effective cold-start solution** for collaborative filtering models. However, in our paper, we primarily focus on semantic mapping and interpretability. We will incorporate this part in the revised version. Below, we present the analysis and experiment. In FACE, when using an AutoEncoder for disentangled mapping, our model simultaneously trains a decoder that generates original collaborative representations from descriptors. We can leverage textual information from the dataset to generate descriptors for cold-start items. These descriptors can then be fed into the well-trained decoder to generate collaborative representations for cold-start items. The same principle applies to cold-start users.
>
> We conducted an item **zero-shot experiment** to illustrate the model's cold-start performance: For a given dataset, 1/5 of the items were designated as cold-start items and excluded during training. During testing, we utilized the textual summary information to allow the LLM to generate descriptors, mimicking the association between textual information and descriptors of other items. The embeddings of these generated descriptors were then fed into the decoder to produce collaborative representations. Item recommendations were then made by calculating the similarity with user representations. We compared our approach against AlphaRec, a model with strong cold-start performance, using the same base model and textual information. The results are presented below:
>
> | Dataset    | Metric |  AlphaRec  |    FACE    |
> | :--------- | :----: | :--------: | :--------: |
> | **Amazon** |  R@5   |   0.0505   | **0.0630** |
> |            |  R@20  |   0.1370   | **0.1395** |
> |            |  N@5   |   0.0456   | **0.0611** |
> |            |  N@20  |   0.0750   | **0.0912** |
> | **Yelp**   |  R@5   | **0.0347** |   0.0342   |
> |            |  R@20  | **0.1116** |   0.1010   |
> |            |  N@5   |   0.0368   | **0.0374** |
> |            |  N@20  |   0.0621   | **0.0658** |
> | **Steam**  |  R@5   |   0.0447   | **0.0496** |
> |            |  R@20  |   0.1287   | **0.1358** |
> |            |  N@5   |   0.0430   | **0.0562** |
> |            |  N@20  |   0.0729   | **0.0838** |
>
> As shown, FACE facilitates a bidirectional mapping between collaborative filtering representations and LLM vocabulary, thereby **empowering traditional collaborative filtering models with cold-start capabilities**. It achieves better performance than AlphaRec, which relies on a mapping to initial collaborative representations. This demonstrates FACE's strong cold-start and scalability potential.
>
> > Response to W2: baselines & citations
>
> Thank you for your suggestions. Our primary focus, as stated in the Limitations section, is on **collaborative filtering**, a mainstream and classic domain in recommendation, rather than sequential recommendation (e.g., TIGER (Rajput et al. 2023), S3-Rec (Zhou et al. 2020), SASRec (Kang et al. 2018)). Sequential recommendation operates under its own distinct problem settings. However, it's worth noting that our framework could be adapted for sequential recommendation tasks, for instance, by enhancing item embeddings of a sequential recommendation model or by leveraging LLMs for sequential modeling using the mapped descriptors that LLMs can understand, as discussed in Appendix G Broader Impact. We are keen to explore this in future work.
>
> Regarding "SPAE: Semantic Pyramid AutoEncoder for Multimodal Generation with Frozen LLMs," this work is an image tokenization paper in the image-text multi-modal domain, whereas our work performs disentangled mapping in the collaborative filtering scenario. We will cite this paper and add it to our discussion.
>
> > Response to W3, Q1, Q2, & Q3: Implementation Details
>
> **Q1: Do contrastive loss for users and items separately or together?** We conduct contrastive loss for users and items **together** within a single batch, similar to the approach in CTRL (Li et al. 2023), which leads to stable and robust learning during experiments. Performing the loss separately is also a valid approach.  We will add this detail to the paper in revision.
>
> **Q2.1: What is recommendation objective?** Our proposed framework is a **model-agnostic plugin**. Therefore, the recommendation objective is that of the base model. In our experiments, the collaborative filtering base models all adopted the **BPR recommendation loss function as their objective** and were evaluated under the **all-ranking protocol** (calculating similarity between users and all candidate items). These settings are consistent with the original papers to ensure fair comparison.
>
> **Q2.2: Do we optimize all the parameters of the LLM?** In fact, our FACE achieves semantic alignment **without fine-tuning LLMs**, as  stated in our Abstract and Introduction. This is because we semantically map the CF embeddings directly to the LLM's vocabulary, which allows the LLM to understand the representations and leverage its pre-trained capabilities to align without any fine-tuning. Indeed, this represents a significant advantage. Compared to approaches that require fine-tuning, our method demonstrates superior efficiency and generality. We apologize for any misunderstanding regarding LLM fine-tuning, and will emphasize this point more clearly in the next version of the paper.
>
> **Q3: Why Residual Quantization (RQ) if we only consider the first code?** A large body of existing work has demonstrated that Residual Quantization (RQ) exhibits greater stability and superior performance compared to Vector Quantization (VQ). The hierarchical structure of RQ allows for different levels of semantic information to be captured, and summing across layers provides a fine-grained approximation of disentangled embeddings, as discussed in Section 3.2.2,  For interpretability, we used only the descriptors from the first layer in our interpretability experiments because they are most similar to the original disentangled embedding and contains the most salient semantic information. In our experiments, we found that using only the first layer for alignment already yielded excellent results.
>
> > Response to W4: Where the improvement is coming from?
>
> The performance improvement comes from leveraging the pre-trained capabilities of large language models (e.g., world knowledge, reasoning abilities) to effectively align the tokens mapped from collaborative embeddings with textual information in the dataset, thereby injecting textual semantic knowledge. Existing alignment methods (such as RLMRec) perform direct cross-modal alignment, which fails to make good use of semantic logical relationships. Our framework maps collaborative filtering representations to LLM tokens, effectively leveraging the pre-trained knowledge within LLMs. We designed a contrastive alignment loss and employed a series of measures to ensure proper gradient propagation, thereby enhancing model performance through the injection of textual information. Our ablation studies (Section 4.6) demonstrate that the tokenization mapping component alone provides minimal or negligible improvement to the base model (as shown in the "w/o align" part of our ablation experiments). The significant performance gains are observed when the semantic alignment is introduced, proving the effectiveness of our model in the CF representation-to-LLM vocabulary mapping, which enables utilization of LLM knowledge.

---

> ### Author Response · Authors · 2025-08-05
>
> Dear Reviewer,
>
> Thank you for taking the time to review our paper. We have provided detailed responses to address your concerns and questions. Please let us know if there are any remaining points that need clarification - we would be happy to discuss them further.
>
> Best regards,

---

> ### Author Response · Authors · 2025-08-07
>
> Dear Reviewer dB6G,
>
> Thank you for your initial review of our paper. We have provided a detailed rebuttal addressing all your concerns, including:
>
> 1. Clarifying that FACE is an LLM-tuning-free solution
> 2. Providing scalability analysis and cold-start experimental results
> 3. Adding comparisons with additional baselines as requested
> 4. Clarifying implementation details
>
> We would greatly appreciate your feedback on our responses. As the discussion deadline approaches, we look forward to engaging in further dialogue to address any remaining concerns.
>
> Thank you for your time and consideration.
>
> Best regards,

---

> > ### Comment · Reviewer_dB6G · 2025-08-07
> >
> > Thank you for the response and further clarifications. I greatly appreciate the new cold-start experiment that were included to show that this method can be used for cold-start generalization.
> >
> > *Regarding scalability:* I am not sure if a large space of total token combinations imply that this method is scalable. The effective token space can be significantly smaller than the possible theoretical combinations of tokens since the representation depends on how well the codebook is utilized. Under utilization of codebook is a common occurrence in RQ-VAE training. Without empirical evidence, it is hard to say that this method would scale. From where it stands, the experiments are done only for small datasets (max of 11k items as per Appendix), whereas, real recommender systems have millions-billions of items in the corpus.
> >
> > Even though some of my concerns still remain, I think the pros outweigh the cons. Hence, I will increase the rating to 4.

---

> > > ### Author Response · Authors · 2025-08-07
> > >
> > > Dear Reviewer dB6G,
> > >
> > > Thank you very much for your constructive feedback and for raising your rating. We greatly appreciate your recognition of our work.
> > >
> > > We will add the additional cold-start experimental results and include more discussions on scalability in the revised version.
> > > Thank you again for your valuable comments.
> > >
> > > Best regards,

---

### Decision · Program_Chairs · 2025-09-17

**Decision:**

Accept (poster)

**Comment:**

This paper introduces FACE, a framework for mapping collaborative filtering embeddings into discrete, interpretable LLM tokens. The reviewers reached a strong positive consensus, highlighting the practical advantage of its LLM-tuning-free design, and the demonstrated improvements in both performance and interpretability.

However, I find reviewers have missed critical related work the paper fails to discuss, compare, or position itself against. These include recent works, e.g., Demystifying Embedding Spaces [Tennenholtz et al., 2024]; Embedding-Aligned Language Models [Tennenholtz et al., 2025]; Factual and Tailored Recommendation Endorsements [Jeong et al., 2024]; and Item-Language Model [Yang et al., 2024]. These papers also address the critical challenge of making latent recommendation embeddings interpretable or usable by LLMs, albeit through different methods like direct embedding injection via adapters or reinforcement learning. This omission significantly weakens the paper's claims of novelty in addressing this general problem and reflects an incomplete survey of the immediate research landscape.

While this is a serious flaw, the core technical method in FACE—using a quantized autoencoder to map embeddings to a set of existing, human-readable LLM tokens—remains distinct from the approaches in the uncited literature. Given this technical distinction and the unanimous post-rebuttal support from all four reviewers, the paper is considered borderline. The final recommendation is to accept, contingent on the authors performing a revision of the introduction and related work sections. The camera-ready version must thoroughly discuss the concurrent literature, clearly articulate the unique contributions of FACE's mapping-based approach, and properly contextualize its place within the broader effort to bridge the gap between recommender systems and LLMs.